# Empowering World Models with Reflection for Embodied Video Prediction

Xiaowei Chi [* 1]  Chun-Kai Fan [* 2]  Hengyuan Zhang [* 2]  Xingqun Qi [1]  Rongyu Zhang [2]  Anthony Chen [2]
Chi-Min Chan [1]  Wei Xue [1]  Qifeng Liu [1]  Shanghang Zhang [✉ 2]  Yike Guo [✉ 1]

## Abstract

Video generation models have made significant progress in simulating future states, showcasing their potential as world simulators in embodied scenarios. However, existing models often lack robust understanding, limiting their ability to perform multi-step predictions or handle Out-of-Distribution (OOD) scenarios. To address this challenge, we propose the Reflection of Generation (RoG), a set of intermediate reasoning strategies designed to enhance video prediction. It leverages the complementary strengths of pre-trained vision-language and video generation models, enabling them to function as a world model in embodied scenarios. To support RoG, we introduce Embodied Video Anticipation Benchmark(EVA-Bench), a comprehensive benchmark that evaluates embodied world models across diverse tasks and scenarios, utilizing both in-domain and OOD datasets. Building on this foundation, we devise a world model, Embodied Video Anticipator (EVA), that follows a multistage training paradigm to generate high-fidelity video frames and apply an autoregressive strategy to enable adaptive generalization for longer video sequences. Extensive experiments demonstrate the efficacy of EVA in various downstream tasks like video generation and robotics, thereby paving the way for large-scale pre-trained models in real-world video prediction applications. The video demos are available at https://sites.google.com/view/icml-eva.

*Equal contribution . The research was supported by Theme-based Research Scheme (T45- 205/21-N) from Hong Kong RGC, NSFC (No. 62206234), the National Natural Science Foundation of China (62476011), and Generative AI Research and Development Centre from InnoHK. [1]Hong Kong University of Science and Technology [2]State Key Laboratory of Multimedia Information Processing, School of Computer Science, Peking University. Correspondence to: Shanghang Zhang <shanghang@pku.edu.cn>, Yike Guo <yikeguo@ust.hk>.

*Proceedings of the 42nd International Conference on Machine Learning*, Vancouver, Canada. PMLR 267, 2025. Copyright 2025 by the author(s).

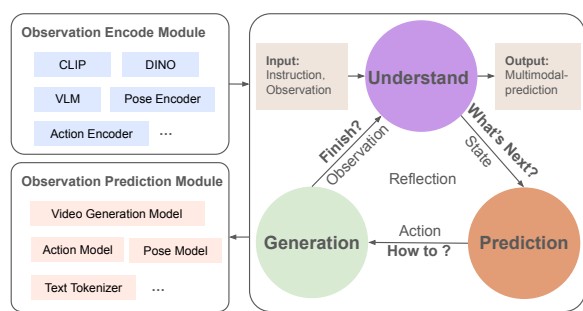

*Figure 1.* The illustration of the **Reflection of Generation(RoG)**. Giving the instruction and observation as input, the world model gives a proper output with the combination of understanding, prediction, and generation.

## 1. Introduction

The rapid development of video generation technologies has brought predictive models to the forefront of building embodied world simulators (Agarwal et al., 2025), where the ability to anticipate future status is critical in embodied scenarios (Wang et al., 2024b). These generated predictive videos can serve as interactive guidelines, mixed-reality product manuals, driving instructors (Gao et al., 2024; Wang et al., 2023b), gaming assistants (Bruce et al., 2024), or even a robot's planning imagination (Du et al., 2023a; Yang et al., 2023). By imagining how the world evolves as an agent behaves, this prediction capability significantly enhances decision-making in embodied scenarios, providing a foundation for robust world models (Ha & Schmidhuber, 2018).

However, existing video generation models for robotics (Zhou et al., 2024; Yang et al., 2023) primarily focus on conditional simulation, neglecting the critical role of understanding both current and predicted states. Therefore, these models are limited to generating fixed-frame sequences and single-round predictions. Moreover, they lack the ability to detect flawed outputs, relying heavily on human intervention. These limitations severely degrade their performance in Out-of-Distribution (OOD) scenarios and undermine the fundamental goal of an embodied world simulator.

To address these challenges, we propose Reflection-of-Generation (RoG), inspired by recent advances in reasoning

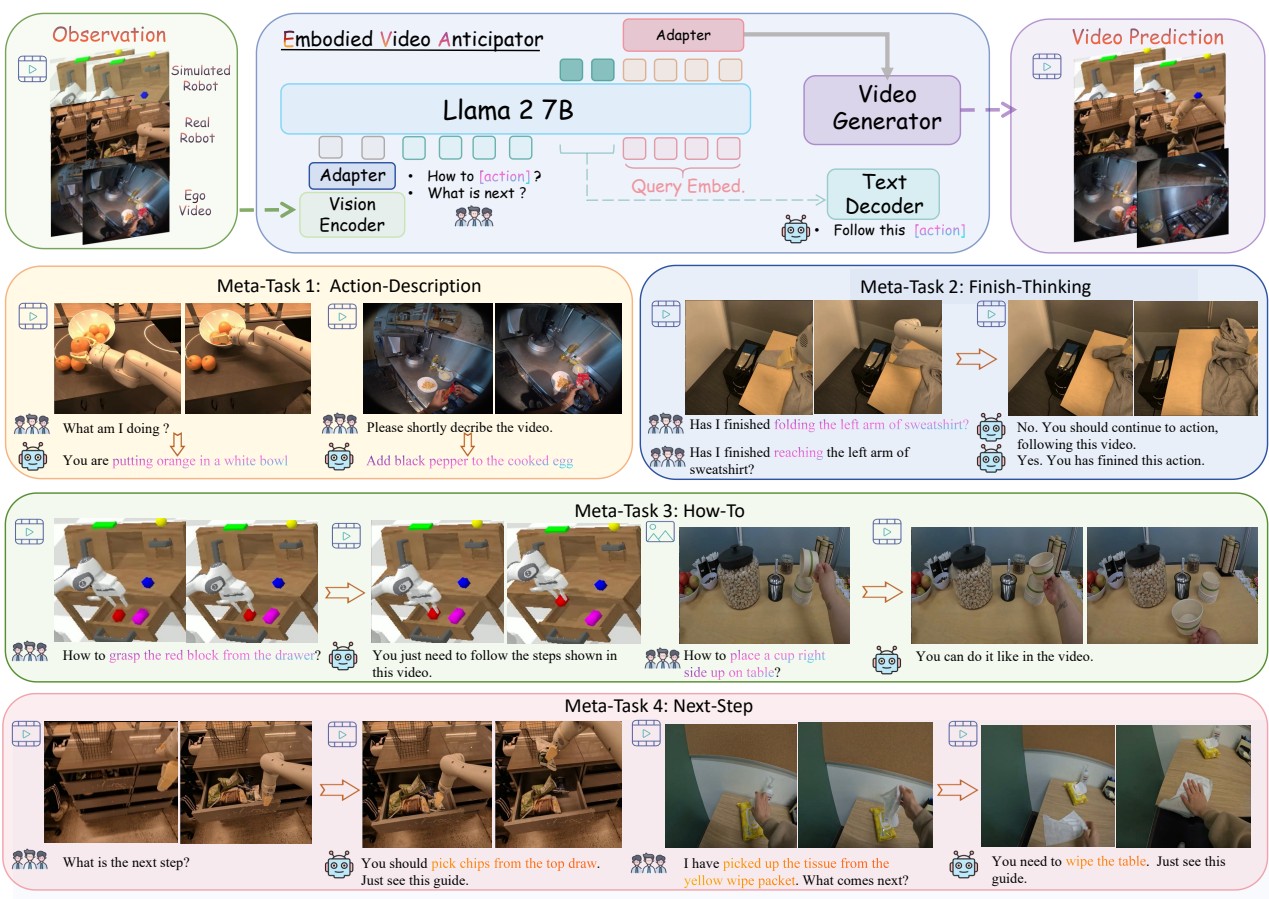

*Figure 2.* **Meta-tasks of the embodied-video prediction.** We present four meta-tasks, including Action-Description, Finish-Thinking, How-To, and Next-Step, for embodied video anticipation and build the related dataset, benchmark, and model.

and alignment (Wei et al., 2022; Liu et al., 2024b; Kawaharazuka et al., 2024). As shown in Figure 1, RoG integrates intermediate reasoning steps into the video generation process, enabling iterative self-correction and deeper understanding. It separates the understanding and generation tasks, allowing for a modular and interpretable design. Therefore, RoG can combine any Visual Language Model (VLM) and Video Generation Model (VDM) into a unified world model. By introducing reflection as a critical step, RoG empowers the world model to iteratively refine its predictions, and generate longer video sequences.

To facilitate the training and evaluation of RoG, we introduce EVA-Bench, a comprehensive benchmark designed for embodied video anticipation. EVA-Bench decompose the RoG within embodied scenes into four meta-tasks: *Action-Description, How-To, Finish-Thinking, Next-Step*, as illustrated in Figure 2. These meta-tasks simplify data collection and evaluation by focusing on both understanding and generation aspects. EVA-Bench includes both in-domain and out-of-distribution videos, leveraging diverse datasets from egocentric and robotics scenarios. It evaluates models using a combination of Video Question-Answering (VQA) metrics and pixel-level video prediction metrics.

Building on this benchmark, we propose EVA (Embodied Video Anticipator), a unified understanding and generation model tailored to the RoG. EVA employs a chunk-wise autoregressive paradigm, integrating reflection conditions to adaptively extend the length of generated video sequences. By supervising the generation process with VLMs, EVA achieves consistent and self-correcting video generation, enhancing the interaction between humans and machines in embodied scenarios.

In summary, our contributions are as follows:

- **Reflection-of-Generation(RoG):** We propose RoG, a novel strategy that integrates intermediate reasoning steps into video generation, enabling self-correction and deeper understanding in embodied scenarios.

- **EVA-Benchmark:** We create EVA-Bench, a benchmark that evaluates world models across diverse embodied tasks with in-domain and OOD scenarios, using both understanding and generation metrics.

- **Embodied Video Aiticipator:** We design EVA, a unified understanding and generation world model that generates longer, more consistent videos through chunk-wise autoregressive reflection.

Extensive experiments on EVA-Bench highlight the strong performance of RoG in both in-domain and OOD tasks. Furthermore, to validate the applicability of EVA in robot planning, we evaluate the model using a robot simulator (Mees et al., 2022; Brohan et al., 2022), demonstrating that EVA and RoG effectively support real-world task execution.

## 2. Related Work

**Video Generation** With the advent of diffusion-based visual generation models, there has been significant progress in extending the capabilities of video generation. For instance, models like VideoCrafter (Chen et al., 2023; 2024) and VideoPoet (Kondratyuk et al., 2023) have demonstrated impressive abilities in generating high-quality video. Moreover, video generation models with image conditions like Dynamicrafter (Xing et al., 2023), Stable Video Diffusion (Blattmann et al., 2023) and Animatediff (Guo et al., 2023) meet impressive generation quality and have already been used in many areas. Such weakness also happens in some long video generation methods (Wang et al., 2023a; Yin et al., 2023). These video-generation models lack reasoning abilities and still struggle with consistency and understanding, so recent works also combine generation models with LLMs (He et al., 2024).

**Embodied World Model** World models aim to provide future predictions based on current observations. This concept has been explored in various domains, including Genie (Bruce et al., 2024), which shows interesting ability in gaming simulation, Vista (Gao et al., 2024; Wang et al., 2023b), etc., in autonomous driving. Video prediction is a special world-model-like task, Seer (Gu et al., 2024), AID (Xing et al., 2024) adapting image-to-video generation model to predict the motion of future frames. Additionally, world models such as RoboDreamer (Zhou et al., 2024), Enerverse (Huang et al., 2025) and AVDC (Ko et al., 2023) have been utilized video generation as robot simulators. Unisim (Yang et al., 2023), for instance, combining pretrained web-scale data with embodied videos expands world models' applications. VLP (Du et al., 2023b) integrated language and video generation models for robot planning but stayed at the concept level. A task-level video predictor is still needed.

## 3. Reflection-of-Generation

In this section, we introduce the RoG by first defining hierarchical levels of visual prediction and then the basic process of the RoG world model.

### 3.1. Hierarchy of Video Prediction

The video prediction task, begin with an initial observation $O_0$ at time step 0 and a task instruction $I$, is categorized into hierarchical levels of complexity: frame-level prediction, task-level prediction, and serial task-level prediction.

**Frame-Level Prediction:** The objective at this level is to generate subsequent video frames $O_t(t \geq 1)$, conditioned on the initial observation $O_0$. This formulates a next-frame prediction problem, where the model approximates the distribution: $P(O_t|O_0, I), \quad t = 1, 2, \ldots$

**Task-Level Prediction:** Task-level prediction focuses on achieving a high-level instruction $I$, such as *"pick up the book"*. The goal is to model the sequence of predictions $\{O_t\}$ required to complete the task: $P(\{O_t\}_{t=1}^T|O_0, I)$ Here, $T$ represents the time horizon necessary to complete the task. This level abstracts away low-level frame transitions, instead emphasizing coherent task execution.

**Serial Task-Level Prediction:** The more complex level involves sequentially completing multiple sub-tasks to fulfill a broader goal. For instance, instruct *"clean the table"*, the model must decompose it into sub-tasks (e.g., *"remove objects"*, *"wipe the surface"*) and execute them sequentially.

**Task-Level Prediction via a Unified World Model:** This paper focuses on task-level prediction as the core of the video prediction task, bridging frame-level generation and task reasoning. The central objective of task-level prediction is to simulate the state of the world, which is the role of the world model, denoted by $\mathcal{W}$. Formally:

$$\mathcal{W}(O_0, I) \approx P(\{O_t\}_{t=1}^T|O_0, I) \tag{1}$$

The world model processes $O_0$ and $I$, dynamically combining frame generation and task reasoning to generate the sequence $\{O_t\}$. The predictions are refined iteratively until the task requirements are met. This hierarchical approach ensures that task-level prediction incorporates both low-level frame synthesis and high-level task satisfaction, addressing the demands of complex video prediction tasks.

### 3.2. Reflection of Generation World Model

The RoG world model introduces a dynamic feedback mechanism that integrates an understanding module and a generation module to iteratively refine task-level predictions, as shown in Figure 3. The model is defined as follows:

$$\mathcal{W}_{\text{RoG}}(O_0, I) = \text{Reflect}(H) \cdot P(\{O_t\}_{t=1}^T|O_0, I, H) \tag{2}$$

Where $\{O_t\}_{t=1}^T$ is a sequence of predicted observations, $H$ is the encoded hidden state representing the prediction status, $\text{Reflect}()$ is the understanding module, which provides three types of responses: extend, regenerate, or output. The whole inference process is as shown in Algorithm 1. The algorithm iteratively refines a prediction sequence $\{O_t\}$ using a reflection mechanism. It begins by encoding the initial

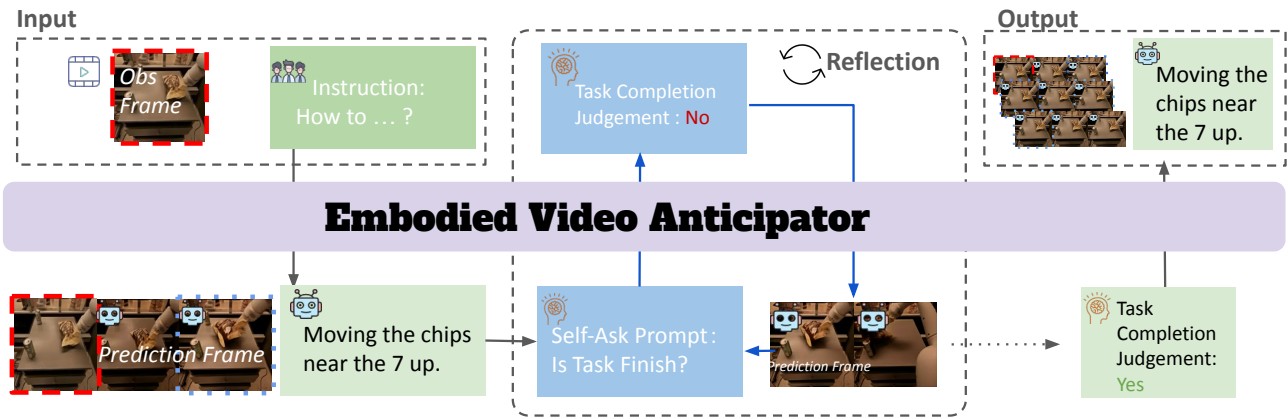

*Figure 3.* **RoG inference pipeline of EVA with chunk-wise frame extension.** Given the visual observation and human questions as input, EVA would first generate fixed frames of videos and related text answers. Then, the model prompts itself to check the task completion status; if the predicted video is not finished, EVA keeps generating the extended frames until the task completion judgment is true.

---

**Algorithm 1** Reflection of Generation World Model ($\mathcal{W}_{\text{RoG}}$)

---

**Input:** Initial observation $O_0$, task instruction $I$
**Output:** Sequence of predictions $\{O_t\}_{t=1}^T$
**Initialize:** Prediction sequence $\{O_t\} \leftarrow \emptyset$
**repeat**
    $H \leftarrow \text{Encode}(O_0, I, \{O_t\})$  # Understanding Module encodes input states
    $\hat{V} \leftarrow \text{Reflect}(H)$   # Generate reflection output based on $H$
    **if** $\hat{V} = \text{Extend}(\{O_t\})$ **then**
        Extend the prediction sequence $\{O_t\}$
    **else if** $\hat{V} = \text{Regenerate}(\{O_t\})$ **then**
        Regenerate the prediction sequence $\{O_t\}$
    **else if** $\hat{V} = \text{Output}(\{O_t\})$ **then**
        Finalize and output the prediction sequence $\{O_t\}$
        **break**
    **end if**
**until** Prediction sequence converges or breaks

---

observation $O_0$, task instruction $I$, and current predictions into a high-dimensional representation $H$.

Based on $H$, the reflection mechanism determines whether to extend the sequence by generating new frames, regenerate flawed predictions, or finalize the sequence if it is complete. This process continues until the predictions converge or meet the task's stopping criteria, thereby ensuring adaptive and accurate video generation in embodied scenarios.

### 3.3. RoG World Model Task Decomposition

The RoG world model is evaluated through four sub-tasks with corresponding metrics, decomposing video prediction into actionable components for a comprehensive assessment. The tasks are as follows:

**Action-Description:** Evaluate the model's understanding ability by generating concise textual descriptions (subject + verb + object + location + destination). Metrics include text similarity, GPT-4 (OpenAI, 2024) keyword matching, and CLIP (Radford et al., 2021) score for description quality.

**Finish-Thinking:** This task assesses whether the frame-level prediction should be extended to complete the task. Metrics include the accuracy of binary outputs ("Yes"/"No"). We introduce the Goal Completion Estimation (GCE) to compare the generated frame with the ground truth, Fréchet Video Distance (Unterthiner et al., 2018) multiple video generation quality metrics from VBench (Huang et al., 2024).

**How-To:** This task transforms task instructions into visual outputs and text responses, evaluated by video metrics and text metrics. The combined score of language and video forms the EVA-Score for How-To tasks.

**Next-Step:** Predict the next action in video sequences and language instructions. The metrics are the same as for How-To, including the accuracy of the QA at the task level and the correlation between the intermodality between the predicted frames and the ground truth.

## 4. Embodied Video Anticipator

As we formulate the problem as a video prediction task as equation 2, we propose the world model EVA. EVA includes two main pre-trained models for multimodal prediction, uses a multistage training strategy, and uses an Ensemble method for domain-specific LoRA (Hu et al., 2021) for VDM and interaction tokens to achieve this complex visual prediction task. We describe these key elements in detail in the following section.

### 4.1. Model Structure

EVA integrates a 7B VLM backbone and a 1.5B VDM to enable high-quality video generation for diverse tasks. The VLM backbone combines a CLIP (ViT-L/14) visual encoder with the Vicuna-v1.5 language model, using a parameter-free token clustering algorithm to reduce video token overhead. Special interaction tokens, such as $< image >$ and $< IMG_P >$, bridge visual and textual inputs, aligning with task-specific processing. The VDM generates 16-frame videos in 2 seconds by conditioning on image, FPS, and language embeddings, leveraging temporal and spatial transformers pre-trained on large-scale video datasets. To enhance adaptability, the Ensemble-LoRA method trains low-rank adaptation modules for each domain (e.g., human-egocentric, real-robot). It combines them using a task token-controlled gating system, ensuring efficient multi-domain generalization. Additionally, a cross-attention generation adapter aligns VLM hidden features with VDM text embeddings via linear transformation, optimized using diffusion denoising loss for superior video generation quality. Detailed module structure is described in Appendix A.

### 4.2. Multi-stage Training Strategy

To train EVA, we employ a three-stage process. In the first stage, we use COCO (Chen et al., 2015) and a curate subset of CC3M-595K (Sharma et al., 2018) to warm up the visual encoder adapter in VLM. The second stage involves aligning the VLM with an instruction tuning dataset, transforming it into an embodied QA bot using multimodal instruction data from sources open-domain data e.g. VideoChatGPT (Maaz et al., 2024), along with our EVA instruction tuning dataset. In the final stage, we adapt the entire pipeline and train the generation adapter and VDM's Ensemble-LoRA for each task. This stage addresses the diverse visual distribution among EVA-Instruction datasets by tuning the Ensemble-LoRA and adapter to maximize generation quality. We introduce detailed information on training stages is provided in the Appendix A.

### 4.3. Autoregressive Frame Extention

The chunk-wise autoregressive frame extension method enables iterative video generation based on task completion verification. Given an input frame $o_0 \in \mathbb{R}^{(B \times 1 \times C \times H \times W)}$, the model first processes it through the VLM to obtain a hidden language state, which is passed to the generation adapter and used as a condition for the VDM to generate an initial video $v^{(0)} \in \mathbb{R}^{(B \times T \times C \times H \times W)}$. After each generation step, the model employs RoG by prefixed prompts as shown in Figure 3 to verify whether the generated video satisfies the task's completion requirement. If the task is incomplete, the model extracts the last 1 and $t$ frames ($v_{T-t,T-1}^{(k)}$) from the current output and uses them as input for the next

round of generation, producing an extended video $v^{(k+1)}$. This process iterates, concatenating successive predictions $(v^{(0)}, v^{(1)}, \ldots, v^{(k)})$ until the VLM determines that the task is complete. By iteratively refining predictions and focusing only on keyframes for extension, this method ensures coherent and task-aligned video generation.

## 5. Embodied Video Anticipation Benchmark

Under the RoG setting, we divide the complex interleaved multimodal generation task into four sub-tasks. This section describes how we separate the tasks and introduces our proposed dataset and benchmark. The fine-grained description of each metric and benchmark composition is provided in Appendix C.

### 5.1. Dataset Construction

We construct four datasets for embodied tasks. The How-To and Action-Description datasets are built by extending prompts using GPT-4o (OpenAI, 2024) and standardizing annotations into a subject-verb-object format, based on the structure. For the Finish-Think dataset, we use the first 25% of videos to identify unfinished tasks and convert relevant RoboVQA (Sermanet et al., 2024) questions into this format. The Next-Step dataset is created using key step annotations from the Ego-Exo4D (Song et al., 2024) dataset and sequential task annotations from RoboVQA (Sermanet et al., 2024), focusing on ordered steps for next-step predictions. The complete EVA instruction tuning dataset comprises 500K QA pairs. Detailed information, including prompt structures, dataset ratios, and data examples, is provided in Appendix C and anonymous supplementary pages.

### 5.2. Benchmark Construction

To ensure comprehensive evaluation, EVA-Bench includes 125 meticulously curated high-quality samples from the datasets introduced above, spanning diverse domains such as real-world robots, simulated robots, and egocentric human daily activities. These samples cover a wide range of scenarios, including pick-and-place tasks, cooking, bike repair, COVID testing, and indoor organization, providing a balanced distribution of tasks. For Finish-Thinking frames, we manually annotate the task completion frames. For out-of-distribution (OOD) evaluation in robotics, experts carefully annotate the prompts. Additionally, we rigorously annotate and refine all 125 samples and their corresponding prompts to ensure accuracy and consistency, making EVA-Bench a robust benchmark for embodied video anticipation.

## 6. Experiments

In this section, we give a comprehensive experiment to evaluate the multimodal understanding and generation abil-

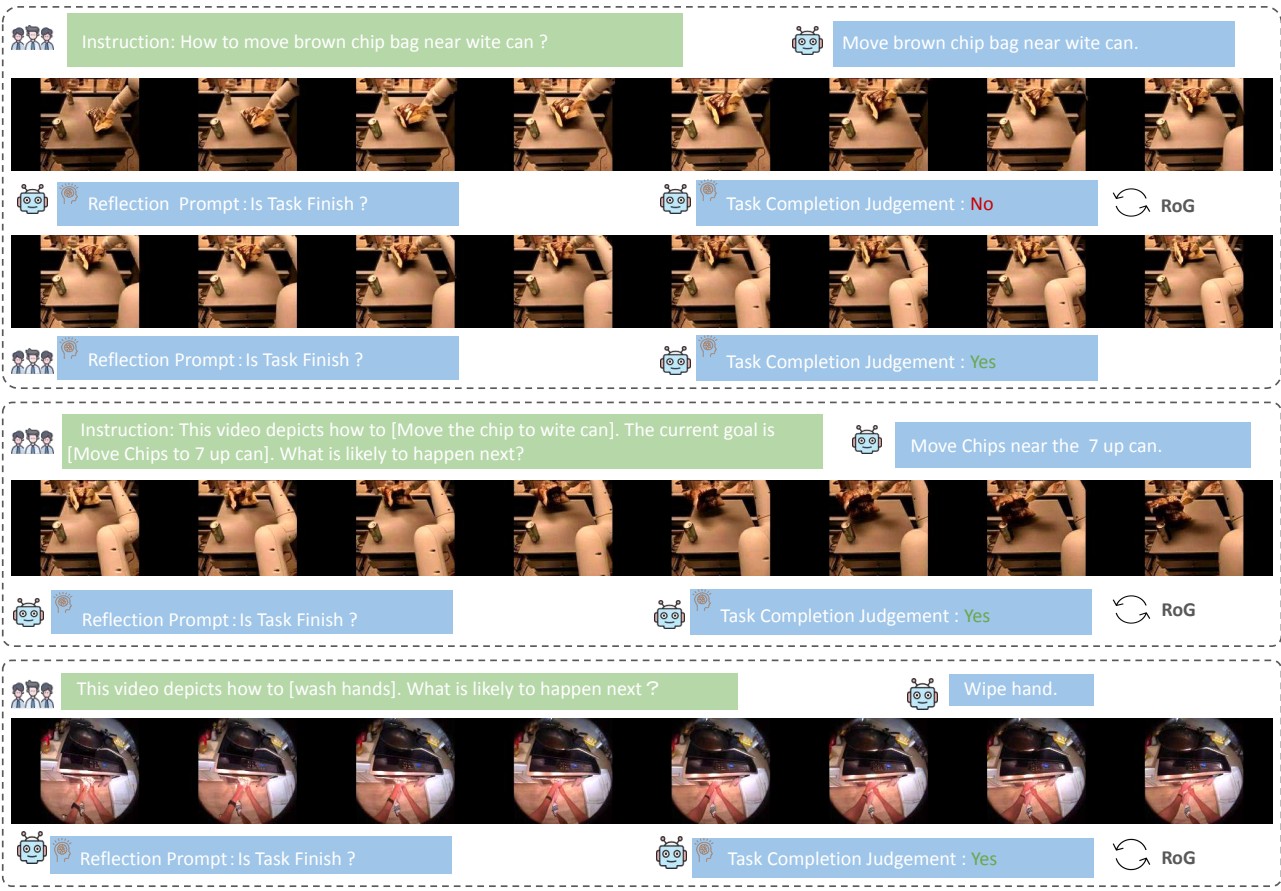

*Figure 4.* **Visualization results of the How-To, Next-Step, and Finsh-Thinking.** Starting from a random statue, EVA can generate robot motion and human-ego motion according to the instructions. The first two continuous cases show the long-horizon generation ability of EVA; in the last example, EVA can generate video based on its reasoning results. We include more example results on the demo page and in the Appendix B.

ity of EVA on four meta-tasks. First, we assess EVA's VQA reasoning abilities on the Action-Description task in Section 6.1, highlighting the knowledge gaps in existing VLMs regarding embodied scenes. Next, we provide the comparison of generation performance in Section 6.2. In Sections 6.3 we compare EVA against four baselines using EVA-Score to represent the capabilities when facing How-To and Next-Step tasks. Last, we demonstrate how the world model performs in robot tasks in 6.4. Both qualitative and quantitative results are presented throughout, with additional qualitative analyses available in Appendix B and page https://sites.google.com/view/icml-eva..

**EVA Instruction Tuning Dataset(EVA-Instruct)** It encompasses four tasks featuring videos of humans, real-world robots, and simulated robots. The complete EVA instruction-tuning dataset consists of 500K QA pairs sourced from Open-X-Embodiment (Padalkar et al., 2023), Ego4d (Grauman et al., 2022), Ego-Exo4d (Grauman et al., 2024), and CALVIN (Mees et al., 2022). The EVA-Instruct is presented in a conversational format, paired with single images and

videos as visual input. Data sources are summarized and more details are included in Appendix A.7.

**Datasets for Evaluation.** EVA-Bench includes a curated collection of 125 high-quality samples from our EVA-Instruct dataset. These samples encompass real-world robots, simulated robots, and egocentric human daily activities. The benchmark is categorized based on meta-tasks. details are included in Appendix C.

**Implementation details.** We set up two kinds of models, EVA-Generator and EVA. EVA-Generator uses Dynami-crafter as the backbone and fully fine-tunes it on the EVA-Instruct. EVA constructs an end-to-end pipeline with fine-tuned ChatUniVi as our VLM backbone and EVA-Generator as our VDM backbone, in the middle, we use a generation adapter to align the feature embedding among two back-bones and only train the adapter with the EVA-Instruct. We also implemented a discreet version of EVA-2stage by using the VLM and VDM in EVA without adapters.

*Table 1.* **Action-Description results in comparison of VLM.** We compare the open-source VLM models. The QA prompts for each model are included in the appendix. In this table, blue means the best-untrained model.

| Model | In-Domain | BLUE-1↑ | METEOR↑ | R-L↑ | CIDEr↑ | Spice↑ | CLIP↓ | GPT-4o↑ |
|---|---|---|---|---|---|---|---|---|
| ChatUniVi (Jin et al., 2024) | ✗ | 0.0969 | 0.0640 | 0.1497 | 0.0427 | 0.0636 | 27.49 | 9.03 |
| LLAVA-NI (Li et al., 2024b) | ✗ | 0.0725 | 0.0741 | 0.1174 | 0.0843 | 0.0982 | 29.63 | 26.94 |
| LLAVA-NV (Zhang et al., 2024c) | ✗ | 0.0717 | 0.0642 | 0.1062 | 0.1267 | 0.0961 | 30.36 | 25.56 |
| LLAVA-O (Li et al., 2024a) | ✗ | 0.0874 | 0.0591 | 0.1118 | 0.2172 | 0.1043 | 27.97 | 22.35 |
| Minicpmv2 (Hu et al., 2024) | ✗ | 0.0672 | 0.0572 | 0.0913 | 0.0404 | 0.0456 | 28.88 | 17.63 |
| Qwen2 (Yang et al., 2024) | ✗ | 0.2484 | 0.1434 | 0.3255 | 0.8914 | 0.2839 | 28.98 | 29.58 |
| GPT-4o (OpenAI, 2024) | ✗ | 0.2651 | 0.1671 | 0.2902 | 0.7355 | 0.3015 | **22.96** | 33.19 |
| ChatUniVi-loRA | ✓ | 0.3007 | 0.1054 | 0.3268 | 0.8245 | 0.2213 | 24.89 | 31.94 |
| ChatUniVi-FP | ✓ | 0.4105 | 0.1809 | 0.4416 | 1.9012 | 0.3414 | 25.36 | 38.46 |
| EVA | ✓ | **0.5735** | **0.3095** | **0.5873** | **4.0139** | **0.5506** | 24.98 | **62.63** |

## 6.1. Video Question Answering

**EVA-Language Metrics.** First, we present the QA task results from our EVA-Bench. Comparing the BLEU (Papineni et al., 2002), METEOR (Banerjee & Lavie, 2005), ROUGE-L (Lin, 2004), CIDEr (Vedantam et al., 2015), SPICE (Anderson et al., 2016), and CLIP (Radford et al., 2021) scores as multimodal measures. Furthermore, to have a better word analysis, we use GPT-4o (OpenAI, 2024) as an automatic evaluator to obtain a GPT-4o score. The detailed metric description is included in Appendix A.7.

**Main Results.** In Table 1, we make the model fine-tuning on EVA-instruct as in-momain and compare a serial of zero-shot VLMs. Qwen2 (Yang et al., 2024) and GPT-4o excel in zero-shot inference. Under GPT-4o's evaluation, while Qwen2 remains the best-performing open-source model with a score of 29.58, the gap between the LLaVA-NeXT (Liu et al., 2024a) series and Qwen2 has narrowed.

Fine-tuned models are trained on a 50K subset of our EVA instruction dataset by LoRA, or full-paramater(FP). Notably, our EVA model achieves the best performance with a score of 62.63 under GPT-4o's evaluation, demonstrating that the diverse data and multi-stage training strategy used in EVA surpass fine-tuning on pre-trained weights.

## 6.2. Longer Video Generation

**EVA-Video Metrics.** We use Subject Consistency (SC), Background Consistency (BC), Motion Smoothness (MS) (Huang et al., 2024), Fréchet Video Distance (FVD), and especially Goal Completion Estimation(GCE) for task completion evaluation. The quantitative experiments are separated into three groups, as shown in Table 2. Finish-Thinking also includes VQA accuracy comparison on the output of Yes/No, which is not included in the table.

**Baselines** For the model with only the video generation module, which takes image and text as input, we compare the Dynamicrafter (Xing et al., 2023), its fine-tuned version, and EVA's video generation module(EVA-Gen). For the

*Table 2.* **Finish-Thinking Video Generation Quality Comparison.** Subject Consistency(SC), Background Consistency(BC), Motion Smoothness(MS), Goal Completion Estimation(GCE), Fréchet Video Distance(FVD). Blue means second-best results.

| Model | Input | SC ↑ | BC ↑ | MS ↑ | GCE ↑ | FVD↓ |
|---|---|---|---|---|---|---|
| DC (Xing et al., 2023) | I+T | 87.25 | 91.91 | 96.72 | 80.96 | 362.56 |
| DC-Tune | I+T | 83.49 | 89.70 | 97.87 | 80.72 | 235.52 |
| EVA-Gen | I+T | 95.74 | 95.11 | 99.09 | 86.83 | 177.28 |
| ChatUniVi+DC | V | 87.10 | 90.82 | 96.97 | 80.80 | 314.11 |
| Qwen2+DC | V | 87.13 | 91.39 | 96.29 | 81.96 | 307.33 |
| ChatUniVi+EVA-Gen | V | 96.54 | 95.26 | 99.19 | 88.48 | 189.61 |
| Qwen2+EVA-Gen | V | 96.13 | 95.48 | 99.15 | 85.87 | 193,89 |
| LLAVA-O+EVA-Gen | V | 96.64 | 95.54 | 99.17 | 88.74 | 192.83 |
| EVA-2Stage | V | 96.68 | 95.82 | 99.17 | **90.19** | 185.89 |
| EVA | V | **97.11** | **96.01** | **99.31** | 89.09 | **184.81** |

*Table 3.* **How-To and Next-Step** Task-Level generation evaluation result on the EVA-Bench.

| Task | Model | EVAS-L ↑ | EVAS-V ↑ | EVA-Score ↑ |
|---|---|---|---|---|
| HOW-TO | LLAVA-O+EVA-Gen | 33.81 | 67.41 | 50.61 |
| | Qwen2+EVA-Gen | 41.54 | 69.34 | 55.44 |
| | EVA-2Stage | **85.51** | **73.32** | **79.42** |
| | EVA | 85.51 | 77.83 | 81.67 |
| Next-Step | LLAVA-O+EVA-Gen | 16.75 | 54.19 | 35.47 |
| | Qwen2+EVA-Gen | 42.99 | 60.11 | 51.55 |
| | EVA-2Stage | 73.02 | 64.46 | 68.74 |
| | EVA | **73.02** | **68.68** | **70.85** |

model with the understanding module under the RoG setting, we compared a series of 2-stage VLM+VDM models. We give the video as the input condition and ask the model to reconstruct. The input video is first converted into a text description by the VLM and then recreated using the first frame and the description.

**Main Results** For the video generation model with text and image as input, our fine-tuning version EVA-Gen improves significantly in GCE and FVD, with scores of 86.83 and 177.28, respectively, highlighting the effective LoRA design. Among these frameworks, EVA excels in SC(97.11), BC(96.01), and MS(99.31) and achieves the lowest FVD(184.81) score. This experiment demonstrates the quality of EVA and highlights the efficiency of RoG in task completion rate.

Table 4. Task successful rate on in-domain and OOD tasks on RT-1. We randomly select six tasks from the RT1 validation set, spanning *Pick Object, Move Object Near Object, Place Object Upright, Knock Object Over, Open/Close, Place Object into Receptacle.*

| Domain | Model | Pick | Move to | Place | Knock Over | Open/close | Place into | Sum. |
|--------|-------|------|---------|-------|------------|------------|------------|------|
| In-Domain | AVDC | 11/16 | 9/12 | 1/2 | 4/4 | 2/8 | 2/8 | 29/50 |
| | w/o RoG | 13/16 | 12/12 | 2/2 | 4/4 | 4/8 | 6/8 | 41/50 |
| | **EVA** | **13/16** | **12/12** | **2/2** | **4/4** | **4/8** | **7/8** | **42/50** |
| OOD | AVDC | 2/4 | 1/4 | 1/1 | 1/2 | 1/1 | 1/3 | 7/15 |
| | w/RoG | 2/4 | 1/4 | 0/1 | 2/2 | 0/1 | 0/3 | 5/15 |
| | **EVA** | **3/4** | **4/4** | **1/1** | **2/2** | **1/1** | **1/3** | **12/15** |

## 6.3. Interleaved Generation

Among the two-stage methods, we demonstrate that EVA-L has a positive correlation with EVA-V. Understanding this ability can directly help improve generation quality and showcase the effectiveness of RoG. Our proposed model, EVA, outperformed other approaches under the RoG setting, achieving an EVAS-V score of 77.83 and an EVA-Score of 81.64. Qualitative results showcasing EVA's performance with different prompts are included in Appendix B.

As shown in Table 3, for the quantitative result of the Next-Step task, EVA had the best performance. As a result, it provides a better text (semantic) condition to guide EVA-Generator for improved generation outcomes. In contrast, LLAVA-OneVision (Li et al., 2024a) and Qwen2-VL-7B performed worse in this task compared to the How-To scenario due to their inability to accurately predict the next-step description. This clearly demonstrates the importance of a VLM that is thoroughly trained in embodied scenes for the Embodied World Model. Such deficiency in EVAS-L also affects the generation quality, leading to the lower performance of LLAVA-OneVision(51.75) and Qwen2-VL-7B(57.23) on EVAS-V. This comparison also shows the overall consistency and quality of EVA-S.

## 6.4. Evaluation on Robot Planning

**Evaluation on RT1:** We evaluated the success rates of human evaluation tasks in RT1 (Brohan et al., 2022) using the RoboDreamer (Zhou et al., 2024) framework, comparing AVDC, EVA without RoG, and EVA across in-domain and OOD prompts in Table 4. EVA outperformed AVDC with a 28% higher overall success rate, achieving a 100% success rate in the Move Object task, demonstrating strong prompt-following ability. However, the Open/Close task proved particularly challenging due to cases like "open right fridge door," which involve transparent glass doors. In OOD, the EVA also shows an obvious improvement in successful counts, 5 more success cases than AVDC. The RoG contributes significantly, 7 cases higher than the model without RoG, demonstrating the importance of it's frame extension.

**Evaluation on CALVIN** We selected four tasks in CALVIN same as (Luo & Du, 2024). As shown in Table 5, EVA

Table 5. Successful rate in CALVIN, including human evaluation of video, and task successful rate in simulation environments. The quality demos are included on the demo page.

| eval | model | lightbulb | led | rotate | Open drawer |
|------|-------|-----------|-----|--------|-------------|
| Video | w/o RoG | 12/41 | 43/54 | 98/157 | 18/41 |
| | EVA | 35/41 | 45/54 | 124/157 | 36/41 |
| Simulation | w/o RoG +CMLP | 7/41 | 13/54 | 32/157 | 4/41 |
| | EVA+CMLP | 9/41 | 15/54 | 45/157 | 9/41 |
| | w/o RoG+CMLP2 | 10/41 | 32/54 | 48/157 | 11/41 |
| | **EVA+CMLP2** | **32/41** | **43/54** | **90/157** | **25/41** |

significantly outperforms the baseline (w/o RoG) in both video and simulation evaluations, demonstrating its superior robotic planning ability. On the video level, EVA achieves much higher task completion rates across all tasks, highlighting the effectiveness of RoG. In the simulation, we applied two versions of the video-to-action head, named CMLP and CMLP2. CMLP&CMLP2 uses video prediction and first action status as conditions, outputting 16 frames of actions. CMLP2 uses more tricks including simple action chunks, etc. The generated video The combination of EVA with CMLP2 achieves the best performance, significantly surpassing both the baseline and EVA with standard CMLP. These results confirm that both RoG and the enhanced CMLP2 model contribute to improved task planning and execution, and demonstrate the potential of this approach in robot task planning.

## 6.5. Qualitative analysis

Figures 4 showcase EVA's capabilities in task execution, rethinking, and long-horizon guidance. In one example, EVA generates the motion "Move brown chip bag near the white can," and through the reflection of generation, extends frames until task completion. In an egocentric human QA scenario, EVA responds to a Next-Step question with "Wipe hand" and generates the corresponding video. Further results, including action-following in real and simulated robots, Video QA outputs, and dynamic adaptability in egocentric videos, are detailed in Appendix B. Besides, more video demonstrations are on the anonymous page at https://sites.google.com/view/icml-eva.

## 7. Conclusion

In conclusion, our proposed Reflection of Generation (RoG) stretagy and the Embodied Video Anticipator (EVA) model demonstrate significant advancements in video prediction for embodied scenarios. By integrating video generation with reasoning tasks like VQA, and leveraging a chunk-wise auto-regressive strategy for longer sequences, EVA achieves high-fidelity predictions and robust generalization. The introduction of EVA-Bench further enables comprehensive and fine-grained evaluation, validating the effectiveness of RoG across diverse downstream tasks. These contributions

pave the way for applying large-scale pre-trained models to real-world video prediction and embodied AI applications.

## Impact Statement

Our EVA and EVA-Bench provide a unified evaluation benchmark to conduct a comprehensive assessment of the World Model's capabilities, and advance the development of World Models in Embodied Intelligence. There are many potential societal consequences of our work, none of which we feel must be specifically highlighted here.

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

# A. Appendix: Model Architecture and Training

EVA enables the pre-trained diffusion generator and visual language model to provide an autoregressive world prediction model.

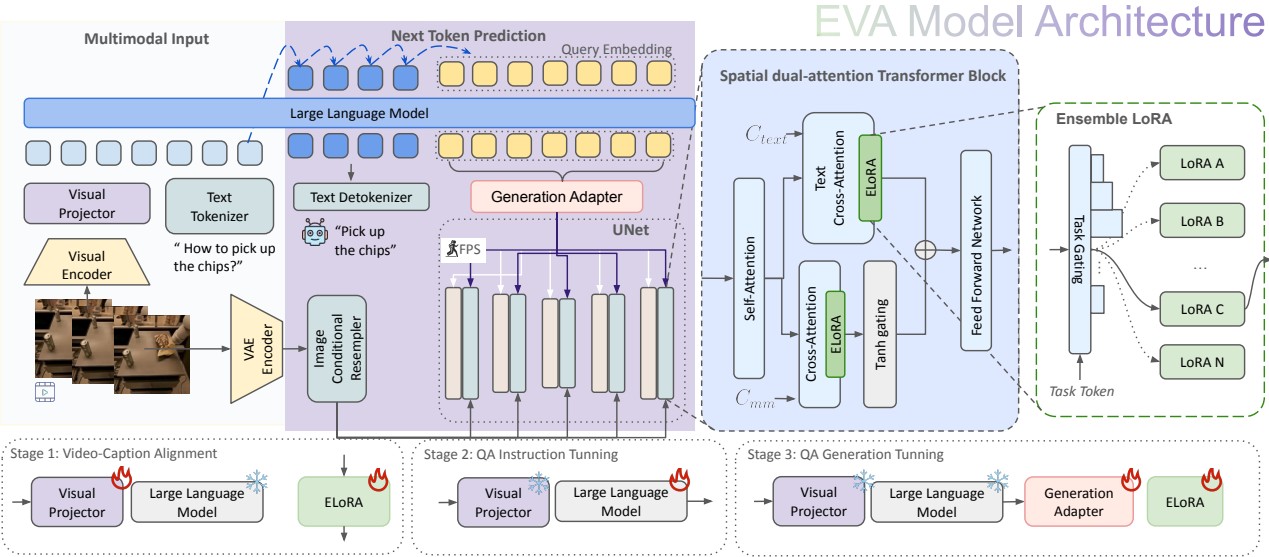

*Figure 5.* **A unified visual understanding and generation framework of EVA.** The EVA introduces a visual projector in understanding LLM, an image conditional resampler in the generation model, trained a generation adapter as a text condition for denoising UNet, and added an Ensemble LoRA system for domain-specific generation. We train the EVA separately, including three stages of alignment and training.

## A.1. Vision Language Model

The Modern Video Language Model (VLM) is based on a Large Language Model (LLM). It leverages the powerful language capabilities of a pre-trained language model to transform the input image or video $I_n$ into latent visual features $\phi$ and then generates a language description output $\tau$. The representation equation of VLM is:

$$\tau = VLM(\phi + \psi) \tag{3}$$

where $\phi = Encoder(I_n)$, and $\psi$ represents the text embedding. Here, $\tau$ is typically in the language embedding sequences, and the visual encoder projects the visual content $I_n$ into the text embedding domain. By this method, VLM trains the visual information in an autoregressive format, similar to another language model.

Given our limited computational resources, it was essential to represent more video frames using fewer video tokens. ChatUniVi's adaptive parameter-free token clustering method significantly reduces the number of video tokens without introducing additional computational overhead, allowing training to remain within the 2048-token limit. We independently trained the VLM on the Embodied-Video-Description dataset, as shown in Fig. 5.

We tested several VLM backbones, including ChatUniVi (Jin et al., 2024), LLaVA-OneVision (Li et al., 2024a), LLaVA-NeXT (Liu et al., 2024a), MiniCPM (Hu et al., 2024), and other mdoels. We found that while existing models were capable of generating detailed descriptions, they struggled with tasks involving prediction and planning, primarily due to the lack of relevant data in their training corpora. Additionally, the simplicity of the text prompts used in the diffusion model's training data necessitated concise responses from the VLM. These responses needed to be composed of short, straightforward sentences that clearly include a subject, verb, object, location, and destination. As a result, we needed to fine-tune the VLMs fully on our dataset.

## A.2. Latent Diffusion Model

The Diffusion Model (Ho et al., 2020) is a type of generative model that iteratively refines a noisy input to generate high-quality data samples. It leverages a series of denoising steps to transform an initial noise distribution $\mathbf{z}_0 \sim \mathcal{N}(0, \mathbf{I})$ into a desired data distribution $\mathbf{x}$. The process involves gradually removing noise from the input, guided by a learned model $\epsilon_\theta$ to produce a clear and coherent output. The latent diffusion model(LDM) further uses VAE to scale down the input features and reduce computation costs. Given the initial condition, the representation equation of the LDM (Rombach et al., 2022) is:

$$\mathbf{x}_T = VAE(Unet(\mathbf{z}_0) + \epsilon) \tag{4}$$

where $\mathbf{z}_0$ is the initial noise, $\mathbf{x}_T$ is the final output, *Unet* is the denoising network, and $\epsilon$ represents the text or other control condition embeddings. The Variational Autoencoder (VAE) further decodes the latent feature into video.

In our comparisons with Animatediff (Guo et al., 2023), VideoCrafter2 (Chen et al., 2024), and Open-Sora (Zheng et al., 2024), we found that Dynamicrafter (Xing et al., 2023), which employs additional image condition injection methods, excels in retaining low-level features and maintaining high consistency for training-free longer video extensions. The core components of this VDM include a VAE encoder and decoder, an image condition resampler, and a denoising UNet.

**Interaction Token** Inspired by (Peng et al., 2023; Dong et al., 2024), we use special tokens to fit the task better. For image input, the VLM uses an `<image>` token as a placeholder within text tokens. Before being processed by LLM, the `<image>` token is replaced by visual feature tokens, obtained through a visual encoder and visual projector layer. For video inputs, the number of `<image>` tokens used corresponds directly to the number of frames in the video. During generation, we concatenate prefix token `<IMG_P>` with VLM language input as query embeddings to extract the feature. As shown in Figure 5, after obtaining the query embeddings, we use it as a condition for VDM to substitute for text prompt.

## A.3. Ensemble of LoRA

We propose Ensemble-LoRA, a method designed to adapt to various domains by utilizing distinct LoRA modules. This approach ensures the highest generation quality in multiple robotic and egocentric environments while maintaining the generalization capabilities of the pre-trained Variational Diffusion Model. Let $W$ represent the original weight matrices in the transformer layers. For each domain $d$, we train a low-rank adaptation:

$$W_d = W + \Delta W_d = W + A_d B_d^T \tag{5}$$

where $A_d$ and $B_d$ are low-rank matrices specific to domain $d$. We applied LoRA after each transformer block in video diffusion to quickly adapt the video generation model to different tasks, as shown in Figure 5. Moreover, inspired by the Mixture of Experts (Jacobs et al., 1991), we proposed an Ensamble-LoRA for each domain by a *Task Token* controlled gating system (human-egocentric, real-robot, simulation-robot, etc.):

$$g_d = \text{softmax}(f(\text{Task Token})) \tag{6}$$

where $f$ is a function that maps the task token to gating values, and $g_d$ is the gating value for domain $d$. Therefore, the ensemble output for a given task is computed as:

$$\hat{W} = W + \sum_d g_d \Delta W_d \tag{7}$$

This formulation allows for efficient adaptation across different tasks without discarding previously learned adaptations.

## A.4. Cross-Attention Generation Adapter

Our generation adapter employs a cross-attention module to align the VLM's hidden features with the VDM's text embedding features. Specifically, the adapter first applies a linear transformation to the VLM's output to match the dimensionality of

the VDM's feature space. We use the diffusion denoise loss to train this adapter to achieve the best generation quality. The detailed module information is introduced in Appendix A.

### A.5. EVA Model Architecture

We employ a Vicuna-based VLM and a 3D U-Net architecture VDM to parameterize the EVA model. The model follows the ChatUniVi structure for VLM, and a standard 3D U-Net structure, with a spatial downsampling pass followed by an upsampling pass, utilizing skip connections from the downsampling activations. This process is interleaved with 3D convolution and attention layers. The model and training hyperparameters of EVA are summarized in Table 6 & 7.

*Table 6.* **Model Architecture for EVA.**

| Name | Type | Parameters |
|---|---|---|
| VDM | UNet | 1.4B |
| VAE Encoder | AutoencoderKL | 83.7M |
| Image adapter | Resampler | 48.8M |
| Text adapter | Resampler | 32.3M |
| VLM | ChatUniVi | 7.0B |
| Query embedding | Linear | 262K |

*Table 7.* **Hyperparameters for training EVA diffusion model.**

| Hyperparameter | Value |
|---|---|
| Base channels | 320 |
| Optimizer | Adam ($\beta_1 = 0.9, \beta_2 = 0.999$) |
| Channel multipliers | 1, 2, 4, 4 |
| Learning rate | 0.0001 |
| Blocks per resolution | 2 |
| Batch size | 4 |
| Attention resolutions | 4, 2, 1 |
| Num attention heads | 64 |
| Conditioning embedding dimension | 4096 |
| Conditioning embedding MLP layers | 4 |
| Conditioning token length | 64 |
| Dropout | 0.1 |
| Training hardware | 8 Nvidia A800 chips |
| Training steps | 20000 |
| Diffusion noise schedule | cosine |
| Noise schedule log SNR range | [-20, 20] |
| Sampling timesteps | 50 |
| Sampling log-variance interpolation | $\gamma = 0.1$ |
| Weight decay | 0.0 |
| Prediction target | $\epsilon$ |

### A.6. VLM Training Details

**Stage 1: Video-Caption Alignment.** In the first stage, we train VLM and VDM separately to fit them into the embodied prediction domain. During VLM training, we aim to align the video encoder with a language model using image-caption pairs from various datasets, including COCO (Chen et al., 2015) and a curated subset of CC3M (Sharma et al., 2018) (CC3M-595K) screened by LLaVA (Liu et al., 2023). The Visual Language Model (VLM) is pre-trained for one epoch with a batch size of 128, using the AdamW optimizer (Kingma, 2014) and a cosine learning rate schedule. The learning rate is set to 2e-3, with a warmup rate of 0.03.

**Stage 2: QA Instruction Tuning** In the second stage, we further align the VLM with the instruction tuning dataset,

transforming it into an embodied QA bot. We incorporate open-domain multimodal instruction data from multiple sources: multimodal in-context instruction datasets (MIMIC-IT (Li et al., 2023)), LLaVA visual instruction datasets (Liu et al., 2023), and video instruction data from VideoChatGPT (Maaz et al., 2024). We then add our EVA instruction tuning dataset as described in Section 3. All input images and frames are resized to 336 × 336. This stage is trained for two epochs with a batch size of 128 and a learning rate of 2e-5.

**Stage 3: Adapting the Pipeline and Training Ensamble-LoRA** In the final stage, we adapt the entire pipeline and train the adapter and VDM Ensamble-LoRA specifically for each task, where with the inspiration of the mixture of experts (Jacobs et al., 1991) (Zhang et al., 2024a) (Zhang et al., 2024b). This stage utilizes the EVA instruction tuning dataset for LoRA training. Despite the diverse visual distribution among EVA-Instruction datasets, the language annotations are similar. Therefore, we propose a special tuning method that tunes the Ensamble-LoRA and adapter to maximize generation quality. In this setup, the generation adapter is trained on the whole dataset, while the Ensamble-LoRA is updated on each domain separately. We compare tuning VDM LoRA, CDM full-parameter tuning, and a two-stage inference method in Section 6.

### A.7. Training Dataset

To enhance comprehensive understanding, reasoning, planning, and video prediction capabilities in embodied environments, we meticulously curate a comprehensive training dataset including 500K instances, termed EVA-Instruct. This dataset encompasses four tasks, each containing videos of humans, real-world robots, and simulation robots. To enhance the diversity of prompts, we employed ChatGPT-4v (OpenAI, 2024)t to generate question-answer pairs, which were then applied to different tasks. The complete EVA instruction tuning dataset comprises 500K QA pairs collected from Open-X-Embodiment (Padalkar et al., 2023), Ego4d (Grauman et al., 2022), Ego-Exo4d (Grauman et al., 2024), and CALVIN (Mees et al., 2022). The data sources are shown as Table 8. The text in these datasets can be effectively restructured into components such as subject, verb, object, location, and destination, as illustrated in 11, making it highly suitable for our task requirements. The instructions for each meta-task are as follows.

| | Dataset | # Examples | Weight |
|---|---|---|---|
| **Simulation** | CALVIN  (Mees et al., 2022) | 23k | 0.85 |
| **Real Robot** | RoboVQA  (Sermanet et al., 2024) | 800k | 0.1 |
| | RT-1 data  (Brohan et al., 2022) | 70k | 0.5 |
| **Human activities** | Ego4D  (Grauman et al., 2022) | 3.5M | 0.01 |
| | Ego-Exo4D Keystep data  (Grauman et al., 2024) | 21k | 0.9 |

*Table 8.* **Dataset name, number of training examples, and mixture weights used for EVA-Instruct.**

**Instructions for action description**. The list of instructions used to briefly describe the video content is shown in Table 12. They present the same meaning with natural language variance. Given the complexity of scene understanding in embodied environments, we aim to simplify the problem by selectively incorporating guidelines into the prompt with a probability of 50%. These guidelines are generated using GPT-4V shown in Table 13.

**Instructions for How-to, Finish-Think, and Next-Step**. The list of instructions used to construct the "How-to" format generation is shown in Table14. The instructions for the Finish-Think meta-task are shown in Table 15. Considering dataset differences, we constructed "next-step" prompts using the instructions from Table 16.

**Data Construction for EVA Meta-Tasks**. As shown in Table 8, the data sources for our four tasks are constructed using distinct instructions to create datasets for each meta-task. The datasets for the How-To and Action-Description tasks are relatively straightforward. Given that the textual annotations in embodied scene datasets generally follow the structure shown in Table 11, we only needed to extend the prompts using GPT-4o and standardize them into the format of subject, verb, object, location, and destination. Due to the ease of constructing the How-To and Action-Description tasks, we built two large datasets: How-To-200K and Action-Description-200K. For the Finish-Think dataset, our statistical analysis indicates that taking the first 25% of a video provides good examples of unfinished tasks. Additionally, some datasets within RoboVQA already contain questions regarding task completion, allowing for direct conversion. Based on this approach, we constructed the Finish-Think-50K dataset. Finally, for the next-step dataset, we utilized the key step annotations from the Ego-Exo4D dataset. This dataset marks key steps for each segment of a complete video, making it easier to convert into a next-step prediction task. In the Open-X-Embodiment's real robot datasets, such as RoboVQA, which contain long-horizon

task annotations involving a sequence of multiple steps, we converted these sequential steps into next-step tasks by focusing on the ordered steps provided. Using this approach, we constructed the Next-Step-50K dataset.

## B. Appendix: Extensive experiments

In this section, we provided the results of the extent visualization experiment. Figure 6 provides visualization results on simulation robot data, demonstrating that EVA could drive the robot by text instruction. The result shows that with proper action, the following is quite accurate.

Figure 7 and Figure 8 also show the action following generation ability of EVA in real robots. Figure 7 shows EVA could answer the question by generation the video.

Figure 9 is an ego-centric generation example. With proper training, the model has some planning abilities, generating a continuous motion sequence that first turns right and then gets the sugar.

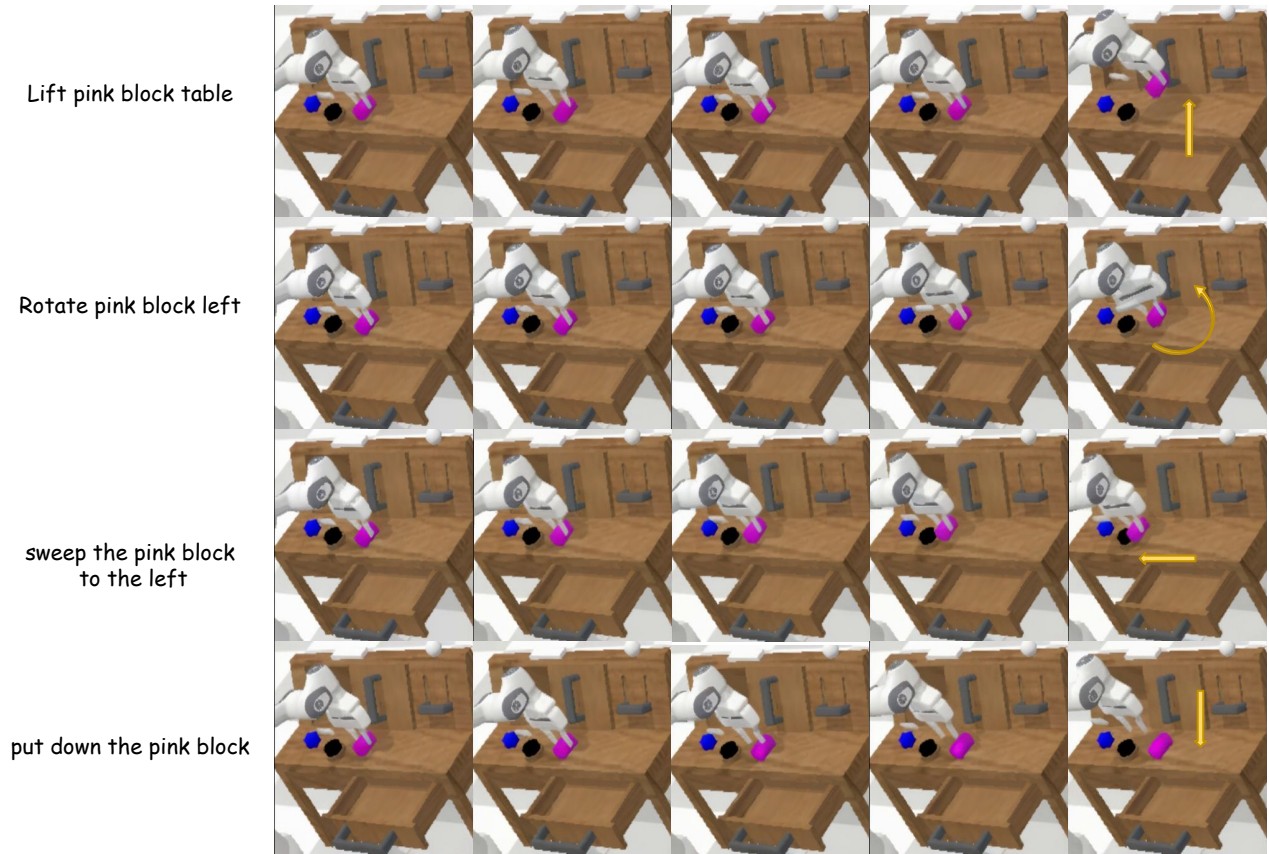

*Figure 6.* We show the prompt following the ability of the EVA on the simulation robot. Given the same input video, the model can generate different actions according to different instructions.

How would you do [pick green can from middle shelf of fridge]?

Could you explain how to [pick green can]?

How to [pick green rice chip bag]?

*Figure 7.* **EVA's action control abilities on the real robot videos.**

## B.1. Comparison in RT1

We compare the success rates of human evaluation tasks in RT1, following the framework of RoboDreamer[6]. Evaluation spans two groups—seen prompts and unseen prompts—and includes a comparison of AVDC, EVA without finish-thinking, and EVA.

| Model | Pick Object | Move Object Near Object | Place Object Upright | Knock Object Over | Open/close | Place Object into Receptacle | Summary |
|---|---|---|---|---|---|---|---|
| AVDC | 11/16 | 9/12 | 1/2 | 4/4 | 2/8 | 2/8 | 29/50 (58%) |
| EVA (w/o finish thinking) | 13/16 | 12/12 | 2/2 | 4/4 | 4/8 | 6/8 | 41/50 (82%) |
| EVA | 13/16 | 12/12 | 2/2 | 4/4 | 4/8 | 7/8 | 42/50 (84%) |

*Table 9.* The human evaluation results of RT-1 (Brohan et al., 2022). We perform human evaluation results to judge the task completion rate on seen prompts and seen cases of EVA and AVDC.

In seen tasks, we randomly selected 50 tasks from the validation set of RT1, including 6 tasks, across multiple scenes(Pick Object, Move Object Near Object, Place Object Upright, Knock Object Over, Open/close, Place Object into Receptacle). For the seen tasks in Tab. 9, AVDC has a 58% success rate, while EVA is 28% higher in total success rate. Moreover, EVA performed better in the Move Object task with a 100% success rate, showing good prompt-following ability. The Open/close task is especially hard since a few cases like "open right fridge door" include the transparent glass door.

| Model/Tasks | Pick Object | Move Object Near Object | Place Object Upright | Knock Object Over | Open/close | Place Object into Receptacle | Summary |
|---|---|---|---|---|---|---|---|
| AVDC | 2/4 | 1/4 | 1/1 | 1/2 | 1/1 | 1/3 | 7/15 |
| EVA (w/o finish thinking) | 2/4 | 1/4 | 0/1 | 2/2 | 0/1 | 0/3 | 5/15 |
| EVA | 3/4 | 4/4 | 1/1 | 2/2 | 1/1 | 1/3 | 12/15 |

*Table 10.* The human evaluation results of RT-1 (Brohan et al., 2022). We perform human evaluation results to judge the task completion rate on EVA and AVDC unseen prompts.

For unseen tasks, we start from the existing cases and manually change the subject or object of the prompt. For example, "Place coke can into the bottom drawer" to "Please close bottom drawer". AVDC performance is better than EVA (w/o finish thinking) in Place, knock, and Open/Close tasks, since the trajectory is longer than EVA (w/o finish thinking) can generate simultaneously. However, EVA could fix this issue and significantly improve the success rate by extending the video.

## C. Appendix: EVA-Benchmark

To facilitate evaluation, the proposed EVA-Bench curated a collection of 125 high-quality samples from our EVA-Instrut, covering real-world robots, simulated robots, and egocentric human daily activities. These samples encompass diverse scenarios such as pick-and-place tasks, cooking, bike repair, COVID testing, and indoor organization. Drawing from existing

knock orange can over

knock redbull can over

move brown chip bag near blue chip bag

move green jalapeno chip bag near apple

move sponge near rxbar blueberry

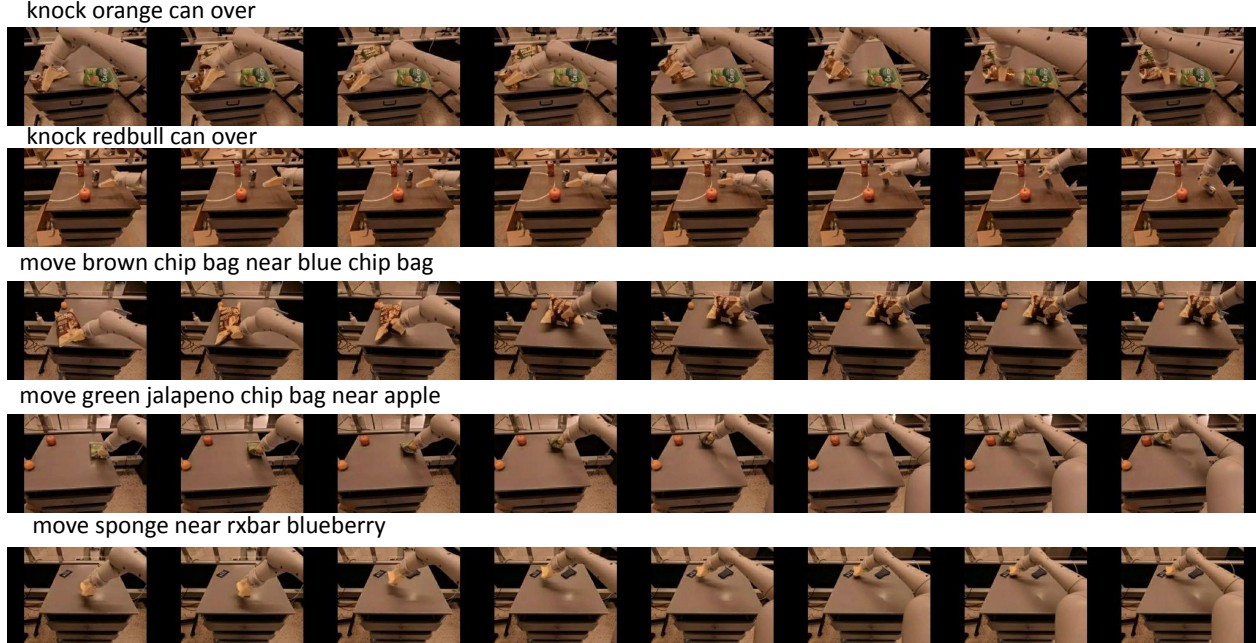

*Figure 8.* **EVA's action control abilities on the real robot videos.**

datasets, we categorize them into three groups: egocentric human videos, real-world robots, and simulations. The statistical distribution of different scenes in our EVA-Bench is shown in Fig. 11.

### C.1. Benchmark Examples

We selected frames from the benchmark in three areas: egocentric human videos, real-world robots, and simulated robots. These frames showcase the richness and diversity of our embodied scenes, as shown in Fig. 10. The first three rows in the figure represent scenes from cooking, COVID testing, and bike repair. The fourth and fifth rows display indoor robotic arms manipulating various objects, while the sixth row shows scenes involving simulated robots.

### C.2. EVA Score Language Metrics

**BLEU (Bilingual Evaluation Understudy Score):** BLEU1 and BLEU2 represent the BLEU scores using 1-gram and 2-gram matches, respectively. BLEU measures the overlap of n-grams between the generated text and reference text. Higher scores indicate greater similarity to the reference.

**METEOR (Metric for Evaluation of Translation with Explicit ORdering):** METEOR considers factors like stemming, synonyms, and word order, making it more flexible than BLEU. It evaluates translation quality based on precision, recall, and a penalty for longer sentences. Higher scores indicate better translation quality.

**ROUGE-L (Recall-Oriented Understudy for Gisting Evaluation - Longest Common Subsequence):** ROUGE-L measures the quality of text summaries and translations by calculating the longest common subsequence (LCS) between the generated and reference texts. It focuses on recall, with higher scores indicating better coverage of the reference content.

**CIDEr (Consensus-based Image Description Evaluation):** CIDEr is used primarily for image description tasks. It evaluates the quality of descriptions by calculating the TF-IDF weighted n-gram similarity between the generated and reference descriptions. Higher scores indicate greater consistency with the reference descriptions. While calculating EVAS-Language points, we normalized the CIDEr by:

$$X_{\text{norm}} = X/10$$

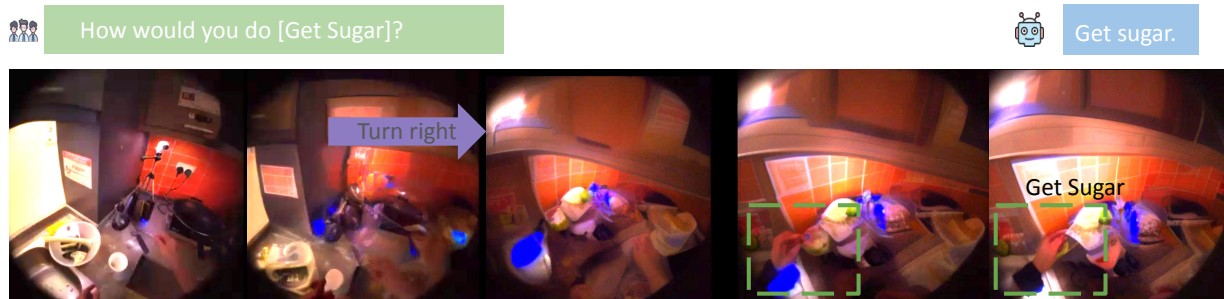

*Figure 9.* **EVA can do large Motion.** In this example, the generation video first turns left and then gets the sugar with a different hand.

where X is the CIDEr score.

**SPICE (Semantic Propositional Image Caption Evaluation):** SPICE evaluates image descriptions by parsing the generated and reference descriptions into semantic graphs. It focuses on the semantic content and relationships within the descriptions. Higher scores indicate better semantic alignment with the reference descriptions.

**CLIP(Contrastive Language-Image Pre-training) score**: CLIPScore is a reference-free evaluation metric for image captioning. Unlike traditional metrics that compare generated captions to reference captions, CLIPScore uses a pre-trained CLIP model to directly measure the similarity between the generated caption and the image itself. This approach leverages the model's ability to understand both images and text, providing a robust evaluation of how well the caption describes the image. Higher CLIPScores indicate better alignment between the image and the generated caption. In EVA Bench, we normalize the CLIP score between 0 1 by:

$$X_{\text{norm}} = \frac{X' - \min(X')}{\max(X') - \min(X')}$$

Where $x'$ is the reciprocal of the $X$, we fix the upper and lower bound by summary the general testing result of different methods.

**GPT-4o as a Judge**: Unlike traditional similarity-based methods, GPT-4o emphasizes semantic understanding. In our implementation, we format the question, model output, and reference into a prompt, as outlined in Appendix C.5, and input it into the GPT-4o evaluator. The comparison between the generated and reference answers is based on four key criteria: object, action type, location, and attribute. During the model evaluation, we observed that some generated responses, despite being semantically close to the ground truth, received low scores. Conversely, responses omitting key information occasionally received high scores. For example, the ground truth might state, "Cut out the tomato stem with a knife on the cutting board" while Qwen2-VL-7B (Wang et al., 2024a) predicts, "Chop the tomato with a knife on the cutting board". Despite high BLEU scores, the key difference between removing the stem and chopping the tomato remains significant. Therefore, using GPT-4o as a judge to score QA text pairs and model-generated responses is essential. By leveraging GPT-4o's advanced analytical and reasoning capabilities, we can more accurately evaluate the similarity between generated and reference texts. The specific prompt is detailed in Appendix C.5.

### C.3. EVA Score Video Metrics

Overall Consistency, Motion Smoothness, Background Consistency, and Subject Consistency metrics are inspired by the contributions from the open-source project VBench (Huang et al., 2024).

**Overall Consistency:** We also evaluate the overall consistency between video and text using ViCLIP (Wang et al., 2022), which measures how well the generated video aligns with general text prompts in terms of both semantics and style.

**Motion Smoothness:** While Subject Consistency and Background Consistency focus on the temporal consistency of the appearance, Motion Smoothness evaluates whether the motion in the generated video is smooth and adheres to real-world physical laws. This is assessed using motion priors from a video frame interpolation model.

**Subject Consistency:** This metric assesses the alignment between the subject in the input image and the subject in the

generated video, also using DINO (Caron et al., 2021) features and order-statistics schemes.

**Background Consistency:** This metric evaluates the coherence between the background scene in the input image and the generated video. It utilizes DINO (Caron et al., 2021) features and carefully designed order-statistics schemes.

**Goal Completion Estimation:** In our benchmark, we manually labeled a frame as the goal condition for loop tasks like "wash hands" lack clear ending frames. Robot tasks often have more defined endpoints, we use the last frame. We use DreamSim (Fu et al., 2023), and use an enhanced version based on DINOv2 (Caron et al., 2021), to compare the generated frames with the ground truth frames. A higher GCE indicates that the generated image is closer to the target.

### C.4. Last Frame Comparison Feature Selection

To measure the similarity between the generated frames and the ground truth, we compare DreamSim(Fu et al., 2023), CLIP(Radford et al., 2021), and DINOv2(Caron et al., 2021). We visualize CLIP and DINOv2 features (Fig.12) to analyze perceptual differences in embodied scenes. Specifically:

CLIP Semantic Heatmap (Chefer et al., 2021), which highlights regions relevant to a text prompt, often misfocuses on irrelevant objects and fails to consistently capture task-relevant interactions. This likely occurs due to the absence of embodied data in CLIP's training set. DINOv2, in contrast, excels at capturing fine-grained features, as evidenced by its attention to interactive objects such as robotic grippers. However, the visualization results of DINOv2 features heavily rely on manually adjusted PCA parameters. It often assigns high attention to background objects, and without proper PCA adjustments, the DINOv2 features of the final frames tend to exhibit extremely low variance, making it difficult to discern meaningful differences. In diverse embodied scenarios, ranging from egocentric tasks to robotic interactions, the PCA values must be frequently adjusted manually for different cases, which undermines the generalization ability of this metric in goal completion estimation tasks.

Ultimately, we select a version of DreamSim which is finetuned on DINOv2-based metric, for its improved alignment with human perception. It better aligns with human perceptual priorities by emphasizing foreground objects and semantic content while remaining sensitive to color and layout. This makes DreamSim particularly generalizable and efficient for evaluating goal state estimation in embodied tasks.

Moreover, we analyze the minimum and maximum DreamSim scores in EVA-Bench and use them to normalize DreamSim as our goal completion estimation (GCE) value, where a higher GCE score indicates closer alignment to the target.

### C.5. Model Inference Prompts

Since the annotated answers in our dataset typically focus on key actions, composed of elements such as subject, verb, object, location, and destination, existing general VLMs struggle to generate responses in the same style. They often produce redundant or irrelevant scene descriptions. To address this, we designed specific prompts to guide the visual language models (VLMs) in generating concise, non-redundant answers. The prompt designs for various VLMs are listed in Table 17. For each meta-task in our EVA-Bench, all models, except for our EVA model, use the same prompts listed in Table 17.

### C.6. Evaluation Prompts

To evaluate general Vision-Language Models (VLMs) in a zero-shot setting, results using standard text evaluation metrics are often poor. Through experimental analysis, we found that traditional metrics like BLEU, METEOR, and ROUGE-L do not effectively capture similarities in actions, objects, locations, and other essential factors. Therefore, we follow the approach proposed by EgoThink (Cheng et al., 2024) and employ GPT-4O to assess the predictions of these VLMs.

However, directly using EgoThink's prompts often leads to extreme scores, with many reasonable predictions being rated as 0. We believe this discrepancy stems from the increased complexity of tasks in embodied scenes, which demands a more nuanced and sophisticated evaluation framework.

For the step description generation task, we guide GPT-4o to assess predictions based on four key criteria: **object**, **action type**, **location**, and **attribute**. Each criterion is scored as follows: 1 if fully correct, 0.5 if partially correct or somewhat aligned, and 0 if incorrect. The final score is the average of these four criteria.

**Object** evaluates whether the objects involved in the action are correctly identified. **Action type** leverages GPT-4O's reasoning ability to assess whether the predicted action aligns with the ground truth. For instance, comparing "get the fork

from the table" and "pick up the fork from the table," the use of "get" and "pick up" would result in a score of 0.5. **Location** assesses whether the location where the action is performed is accurately described, along with the broader context. If the action involves movement (e.g., moving an object from one place to another), the evaluation considers whether the starting point, destination, or path is correctly specified. **Attribute** examines whether the attributes of the objects involved (e.g., size, color, state, condition) are described accurately.

Our designed prompts are presented in Table 18.

- "C adds the shredded ginger into the small stainless cowl with his right hand."

- "C steps to the right while holding her head and swaying her hips."

- "C places his right hand around the waist of woman X."

- "Tighten the left axle nut with your left hand."

- "Place the black chip bag in the tray."

- "Grasp the red block from the drawer."

- "Keep the brown potted plants together."

*Table 11.* **The example answers of EVA-Instruct-Answer.** Example answers from EVA-Instruct-Answer. In this context, C refers to the wearer of the egocentric camera, while x represents the other person involved in the interaction.

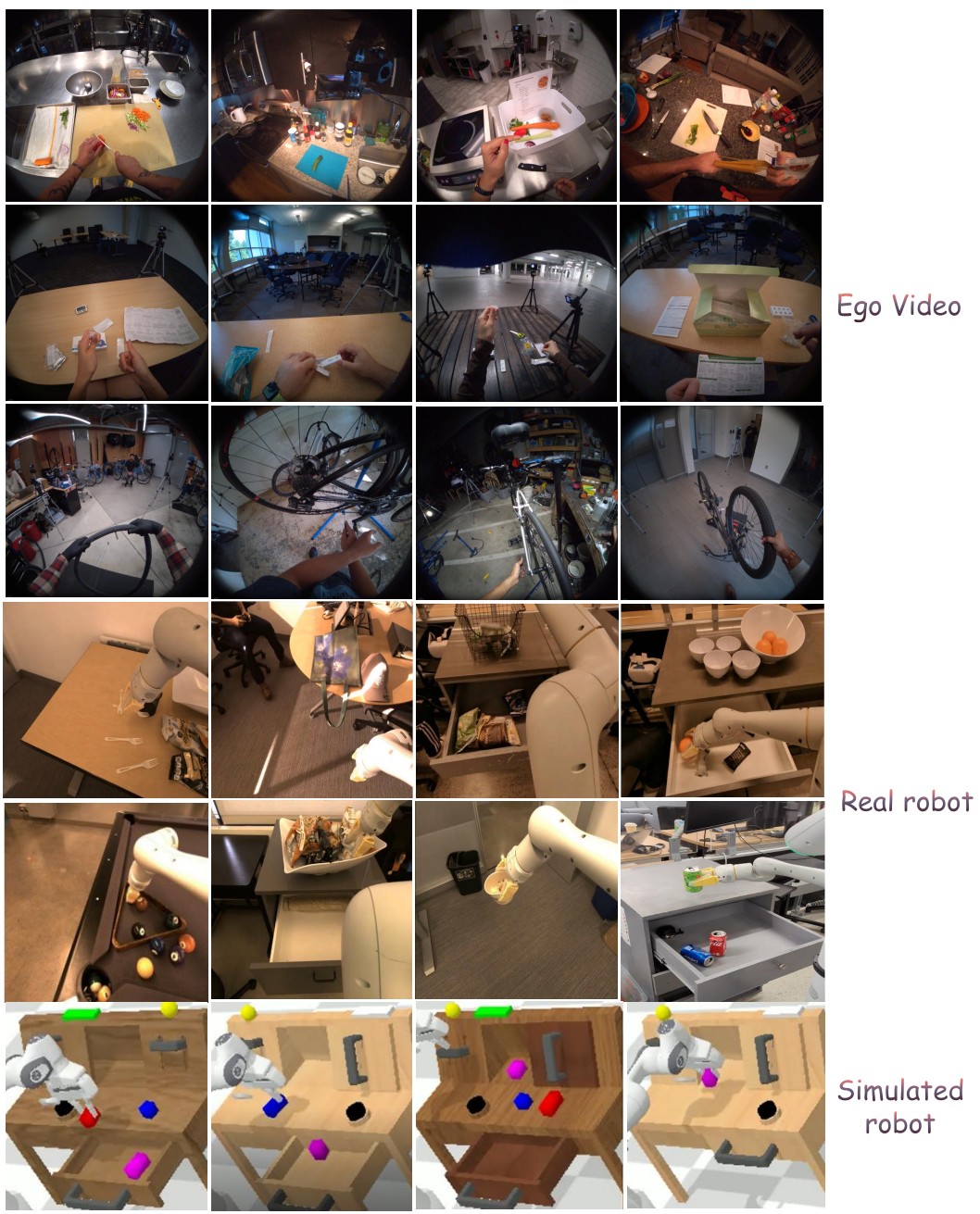

*Figure 10.* **Random frames from EVA-Bench.**

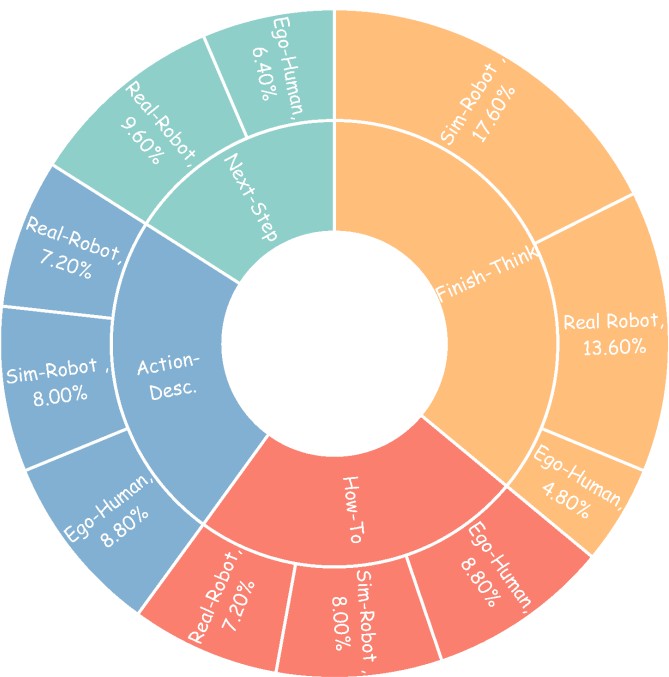

*Figure 11.* This chart illustrates the distribution of various embodied scenes categories within the EVA-Bench.

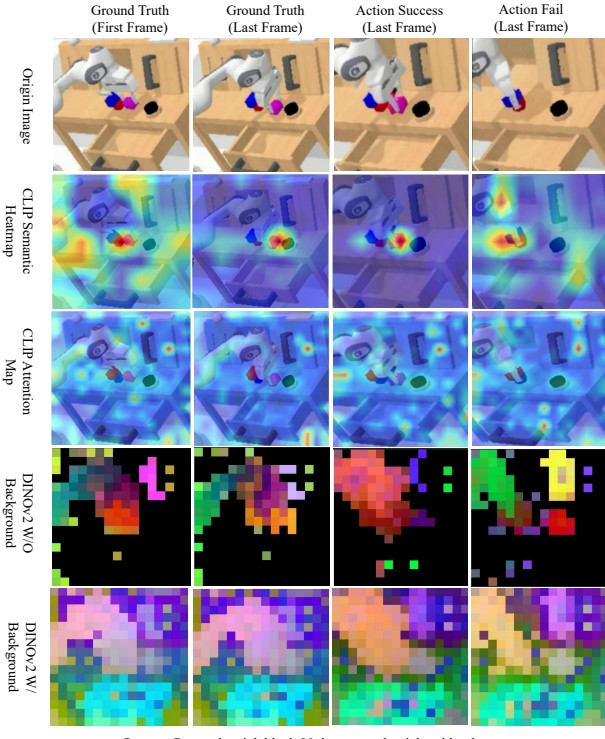

Prompt: Rotate the pink block 90 degrees to the right with robot arm

*Figure 12.* **Visualization of CLIP and DINOv2 Embeddings.** The second row shows the attention heatmaps (relevancy) between the input text prompt and the corresponding image. The third row represents the final attention map from CLIP visual encoder. The fourth and fifth rows illustrate the PCA visualizations of DINOv2 embeddings, showing the top three principal components, with and without background masking, respectively. In the PCA visualizations, brighter regions represent higher values after the PCA transformation.

- "What is happening in this egocentric video?"

- "Can you describe the interaction in this video?"

- "What actions are being performed in this video?"

- "Please provide a description of the activity in this video."

- "What is the person/robot doing in this video?"

- "What object is being interacted with in this video?"

- "Can you summarize the actions in this video?"

- "What task is being carried out in this video?"

- "What is the main activity shown in this video?"

- "Can you provide a brief description of the video content?"

- "What interaction is taking place in this video?"

- "What is the main focus of this video?"

- "What is the subject doing in the video?"

- "Can you explain the actions seen in this video?"

- "What specific steps are being taken in this video?"

- "What is the sequence of actions in this video?"

- "What key activities are being shown in this video?"

- "What is the primary task in this video?"

- "What is the individual engaged in within this video?"

- "What detailed actions are depicted in this video?"

- "Can you outline the main steps in this video?"

- "What is the purpose of the actions in this video?"

- "What are the key interactions in this video?"

- "What process is demonstrated in this video?"

- "What is the sequence of events in this video?"

- "What detailed activities are performed in this video?"

- "What main actions can be observed in this video?"

- "What specific task is being executed in this video?"

- "What are the primary actions taking place in this video?"

- "Can you detail the key steps shown in this video?"

*Table 12.* **The list of instructions for action descriptions.**

- "Please identify the primary object in the first-person view and describe the main actions involving it."

- "Determine the main object and outline the key actions associated with it."

- "First, identify the primary object, then summarize the main actions performed with it."

- "Locate the primary object and describe the key actions taken."

- "Focus on the interaction between objects and the human."

- "Follow these two guidelines: (1) identify the main objects, and (2) describe the key steps."

- "Spot the central item and describe the key steps performed."

- "Point out the central object and explain the key steps involved."

- "Focus on the primary item and narrate the sequence of actions associated with it."

- "Do not confuse the objects or hallucinate about them."

- "Answer based on what you observe, without over-interpreting, distorting facts, or fabricating information."

- "Think like a human: first identify the interacting objects, then infer the actions being performed."

- "First, infer the overall action, then identify the category of objects, and finally, use the object category to determine if the action is correct. Please output the final description of the step."

*Table 13.* **The list of instruction guidelines for action description.**

- "What is the way to [action]?"

- "Can you show me how to [action]?"

- "How can I do [action]?"

- "What is the step to [action]?"

- "Could you explain how to [action]?"

- "What method should I use to [action]?"

- "How should I perform [action]?"

- "What is the best way to [action]?"

- "How would you do [action]?"

- "How to [action]?"

*Table 14.* **List of instructions for How-to meta-task.**

- "The video is describing the step to [action], is this step completed?"

- "Based on the given video, has the task to [action] completed?"

- "Has the action to [action] completed?"

- "Does the video confirm the completion of the step to [action]?"

- "Has the process of [action] been completed based on the given video?"

- "Has the video shown the completion of [action]?"

- "Based on the given video, has the action to [action] completed?"

- "Has the activity of [action] been successfully completed according to the video?"

*Table 15.* **List of instructions for Finish-Think meta-task.**

- "This video depicts how to [action]. The current goal is [goal]. What is likely to happen next?"

- "This video depicts how to [action]. What is likely to happen next?"

- "What is likely to happen next?"

*Table 16.* **List of instructions for Next-Step meta-task.**

| Meta-Task | General Prompts |
|---|---|
| Action-Description | Question: {question}. Please provide a brief description in one sentence. The response should be clear and to the point, containing key action words such as objects, verbs, objects, location, and destination. Avoid unnecessary details or explanations. Please briefly describe the key action in a few words. |
| Finish-Think | This is a " Finish-Think" task where you need to predict if a step is completed or not. Question: {question}. Please answer with either " yes" or " no". |
| How-To | This is a " How-to" task where you need to explain how to accomplish a specific task. Question: {question}. The response should be clear and to the point, containing key action words such as objects, verbs, objects, location, and destination. Avoid unnecessary details or explanations. Please briefly provide a simple step description in a few words. |
| Next-Step | This is a " Next-Step" task where you need to predict the next step in a sequence of steps. Question: {question}. The response should be clear and to the point, containing key action words such as objects, verbs, objects, location, and destination. Avoid unnecessary details or explanations. Please briefly describe the next step in a few words. |

*Table 17.* **Model inference prompts used for four meta-tasks.**

| Model | Prompts for Evaluation |
|---|---|
| GPT-4O | [Instruction] You are tasked with evaluating the quality of the response provided by an AI assistant. The evaluation should focus on **correctness**, **helpfulness**, and **relevance**. Depending on the task type, you will evaluate specific attributes of a step-level generation tasks or score a simple yes/no question.

**1. For step-level generation tasks**, evaluate the assistant's response based on the following attributes:

**Object**: Does the assistant mention the same or a closely aligned object as the reference? Minor but relevant differences (e.g., an additional unnecessary object) can receive partial credit, but introducing unrelated or missing key objects should lower the score.

**Action Type**: Is the action in the assistant's answer precise and in line with the reference? If the intent of the action is similar but less precise, give partial credit. However, if the action significantly changes the task's context or result, it should be more strictly penalized.

**Location**: Does the response correctly identify the location or context of the action? If the action involves movement (e.g., moving an object from one place to another), evaluate if the destination, starting point, or path are accurately described. Minor location discrepancies can receive partial credit, but if the location changes the context or goal of the action, assign a lower score.

**Attribute**: Are the attributes of the object(s) (such as size, color, state, or condition) correctly described? Missing or incorrect key attributes should lead to a lower score. If attributes are implied but still align with the context, partial credit can be given.

**Scoring**: If the reference answer does not include information for a particular attribute (e.g., object, action type, location, or attribute), do not score that attribute. For each attribute, assign:
- 1 if fully correct,
- 0.5 if somewhat correct or partially aligned,
- 0 if incorrect.

After evaluating each attribute, sum the scores and calculate the overall rating by averaging the individual scores. **Do not round the final result**. The final rating will be a non-rounded average score between 0 and 1.

**2. For yes/no questions**, directly evaluate whether the assistant's response is correct: Assign 1 if the answer is correct, and 0 if it is incorrect.

After providing your analysis, rate the response with the calculated average score, formatted as: **"Rating: [[average_score]]"**. Now proceed with the evaluation based on the provided task:

[Task Type] {task_type},
[Question] {question},
[The Start of Reference Answer] {refanswer},
[The End of Reference Answer],
[The Start of Assistant's Answer] {answer},
[The End of Assistant's Answer]. |

*Table 18.* **Model inference prompts used for four meta-tasks.**

