# OpenReview forum: "Empowering World Models with Reflection for Embodied Video Prediction"
_ICML.cc/2025/Conference — ICML 2025 poster_

### Official Review · Reviewer_cisM · 2025-03-10

**Overall Recommendation:** 3

**Summary:**

This paper proposes a world model based on video prediction. The world model is specifically designed for embodied AI (manipulation, more specifically). The authors design a novel strategy, termed Reflection of Generation (RoG), to leverage VLM and video generation models to serve as a world model. Besides, this paper also introduce a benchmark to evaluate embodied world models. Basically this is a good submission, with reasonable methodology and extensive experiments.

**Update after rebuttal**: I appreciate the authors' detailed response. The response has addressed most of my concerns, including the paper organization, video duration and open source. After reading the rebuttal and the comments from other reviewers, I choose to keep my original score, i.e., weak accept.

**Claims And Evidence:**

Yes, the claims are basically supported.

**Essential References Not Discussed:**

I do not find any.

**Experimental Designs Or Analyses:**

Yes, I check the benchmark design.

**Methods And Evaluation Criteria:**

Yes, this paper propose a benchmark for evaluation.

**Other Comments Or Suggestions:**

None

**Other Strengths And Weaknesses:**

Strengths:
1. The RoG method is interesting and novel to me.
2. The inclusion of a benchmark is valuable for evaluating the method's performance.
3. The paper provides comprehensive experimental results, effectively demonstrating the method's effectiveness.

Weaknesses:
1. I'm a little confused by the overall organization of the paper. The benchmark should be presented to evaluate the baseline methods and the proposed method. However, now the benchmark is presented in Sec.4, the basic method is splitted to Sec.3 and Sec.5, which is strange. Besides, Sec.6 describes various details about the evaluation, I'm confused about the difference between this part with Sec.4 (and Appendix C).
2. The video prediction results are a bit short (only 2s for most videos). I'm curious about the results for longer prediction.
3. The video prediction results exhibit noticeable artifacts, e.g., "move brown chip bag near blue chip bag_frame_0_sample0.mp4" in the supplementary.

**Questions For Authors:**

1. Will you open source the code? The supplementary code is "NOT RUNNABLE YET".

**Relation To Broader Scientific Literature:**

The key contributions are related to the world model, which is a hot topic in embodied AI.

**Theoretical Claims:**

Yes, I check the algorithm of the paper, which is reasonable.

---

> ### Author Rebuttal · Authors · 2025-04-01
>
> # Response to Reviewer cisM
> We thank the reviewer for recognizing the novelty of RoG, the value of our benchmark, and the thoroughness of our experimental results. Your constructive feedback helps us further improve the clarity and rigor of the paper.
>
> ---
> ## 1. Paper Organization and Benchmark Presentation
> Thank you for pointing out the confusion regarding the structure of the paper. As our primary contribution lies in the design of a reasoning framework akin to Chain of Thought (CoT) [1], we initially organized the paper to mirror this inspiration. Specifically, we introduced the system framework (RoG) first, followed by the dataset and benchmark (Sec.4), and then the model (Sec.5), to reflect the flow of reasoning.
> However, we agree that the current structure may appear unconventional and potentially confusing. Upon your suggestion, we have revised the organization for better clarity and coherence. In the updated version, we will make the following changes:
> - Move Section 4.1 (Task Decomposition) to the end of Section 3.
> - Swap the order of Section 4 (Benchmark) and Section 5 (Model).
>
> The revised structure is now:
> - RoG Framework: Task Definition, Task Decomposition
> - Model: VLM, LLM, AR Generation
> - Dataset and Benchmark Design
> - Experiments
>
> We believe this reorganization better reflects the logical flow from problem formulation to model design, and then to evaluation. Furthermore, due to the complexity and scale of the foundation models used, we include extensive implementation, fine-tuning, and ablation details in the Appendix.
>
> ---
> ## 2. Short Video Duration in Prediction Results
> Thank you for raising this point. To address this concern, we have provided additional longer video prediction results (8+ seconds) on our anonymous project website under the section "Multi-Round RoG Longer Video Generation": https://sites.google.com/view/icml-eva#h.5u9bchnr2e41
> These extended demos demonstrate the model's ability to perform multi-round robotic interaction tasks over longer horizons. We hope this provides a clearer view of the model's temporal reasoning and generation capabilities.
>
> ---
> ## 3. Visual Artifacts in Generated Videos
> We appreciate your observation regarding visual artifacts. All evaluation cases were randomly selected to ensure fairness and reproducibility. While some artifacts are indeed present, such as in "move brown chip bag near blue chip bag_frame_0_sample0.mp4"—we note that this challenge is not unique to our method. Object consistency in generation, especially deformable objects (e.g., chip bags), remains an open and challenging problem in the video prediction community [2,3].
> Our focus in this work is on generating coherent and temporally consistent robotic behaviors, especially in multi-step tasks. That said, we acknowledge the importance of improving object-level consistency. As part of our future work, we plan to:
> - Introduce spatial consistency constraints during the understanding phase.
> - Integrate additional consistency modules in the video generation phase, particularly for egocentric views (e.g., robot wrist cameras) to better capture fine-grained object interactions.
>
> ## Question: Open Source
> Yes, we plan to open-source both the code and the benchmark data. The current supplementary code is marked as “NOT RUNNABLE YET” because it is still in the experimental stage and tightly coupled with our internal development environment. To ensure broader usability and reproducibility, we are actively working on:
>
> - Cleaning and refactoring the codebase
> -  Packaging the runnable demo
> - Providing environment setup scripts and installation instructions
> - Writing a detailed user manual
>
> We are committed to releasing a public version shortly after the review process, in line with community standards for reproducibility.
>
> We thank the reviewer once again for highlighting this important area for improvement.
>
> ---
> # References
> [1] Wei J, Wang X, Schuurmans D, et al. Chain-of-thought prompting elicits reasoning in large language models[J]. Advances in neural information processing systems, 2022, 35: 24824-24837.
>
> [2] Ren, Weiming, et al. "Consisti2v: Enhancing visual consistency for image-to-video generation." arXiv preprint arXiv:2402.04324 (2024).
>
> [3] Xu, Ziyi, et al. "AnchorCrafter: Animate CyberAnchors Saling Your Products via Human-Object Interacting Video Generation." arXiv preprint arXiv:2411.17383 (2024).

---

> > ### Comment · Reviewer_cisM · 2025-04-02
> >
> > I appreciate the authors' detailed response. The response has addressed most of my concerns, including the paper organization, video duration and open source. I will keep my original score, i.e., weak accept.

---

> > > ### Author Response · Authors · 2025-04-02
> > >
> > > Thank you for your thoughtful review and for highlighting the strengths of our work. Your suggestions have been particularly helpful in improving the clarity and organization of the paper.
> > >
> > > We are revising the manuscript accordingly and working on releasing a clean, runnable version of the code. We truly appreciate your feedback and support — it has definitely helped us improve the quality of our work.
> > >
> > > All authors

---

### Official Review · Reviewer_aEuy · 2025-03-11

**Overall Recommendation:** 3

**Summary:**

In this work, the authors propose a  Reflection of Generation (RoG) solution to enhance video prediction in embodied scenarios. To achieve it, they further introduce an Embodied Video Anticipation Benchmark (EVA-Bench) and an Embodied Video Anticipator (EVA) model.

**Claims And Evidence:**

Most claims are supported by clear evidence.

**Essential References Not Discussed:**

The model style is similar to  [Xiang, et al. Pandora: Towards General World Model with Natural Language Actions and Video States]. Please further make the comparison to show the key difference.

**Experimental Designs Or Analyses:**

The experiments are basically sufficient to support the design.

**Methods And Evaluation Criteria:**

The methods and evalution basically make sense for video prediction within the proposed RoG style.

**Other Comments Or Suggestions:**

Please see the weakness.

**Other Strengths And Weaknesses:**

There are three contributions including RoG solution style, EVA-Bench, and EVA model. However, such style and model has been partially investigated in the [Xiang, et al. Pandora: Towards General World Model with Natural Language Actions and Video States].  This would reduce the importance of RoG and EVA contributions in this paper, especially model structure is similar to Pandora. Please clarify the key differences.

**Questions For Authors:**

Please see the weakness.

**Relation To Broader Scientific Literature:**

The paper provides a possiblity for unifying visual understanding and generation tasks.

**Theoretical Claims:**

No theoretical claims are introduced in the paper.

---

> ### Author Rebuttal · Authors · 2025-04-01
>
> # Rebuttal to Reviewer aEuy
> We sincerely thank the reviewer for the insightful comments and for highlighting the relevance of Pandora. We also appreciate your recognition of the strengths of our work, such as “most claims are supported by clear evidence” and “the methods and evaluation basically make sense for video prediction.” Although both EVA and Pandora share the general goal of integrating vision and language for video generation, we differ substantially in purpose, methodology, and impact. We will explicitly include a comparison with Pandora in the revised manuscript. Below, we provide a point-by-point response to clarify the key differences between EVA and Pandora and why these distinctions are essential.
>
> ## 1. Conceptual Difference (RoG Solution Style vs. Pandora’s Approach):
> The primary contribution of EVA is the Reasoning-over-Generation (RoG) strategy, which introduces a structured thinking paradigm for unified understanding-generation tasks. RoG is analogous to the Chain-of-Thought (CoT) reasoning in LLMs, enabling stepwise reasoning and decision-making in multimodal models. Unlike Pandora, which is primarily a multimodal-to-video generation model, RoG is a generalizable strategic framework that can be applied to any unified understanding-generation model, including Pandora or other LLM+VGM frameworks(Tab. 2 & 3, or we newly add EVA(Qwen) version in https://openreview.net/forum?id=onumui0nHi&noteId=cunTZeC5Vq ). As claimed in our experiments, e.g. Table 2, we demonstrate how RoG enhances reasoning capabilities across different models.
> ## 2. Benchmarking Contribution (EVA-Bench vs. Pandora’s Dataset):
> EVA-Bench is specifically designed to evaluate both visual understanding and generation abilities in embodied scenarios, which include multiple text metrics(Tab.1), video metrics like goal completion estimation(Tab.2, L.1104, Fig.12), and multimodal generation and reasoning metrics(Tab.). Pandora’s dataset, although it includes some robot-related videos, only has a visual evaluation. EVA-Bench incorporates manual caption adjustment, case separation, and task-specific design to ensure fine-grained evaluation. Additionally, in generation evaluation, we introduce goal-conditioned estimation metrics, which are crucial for benchmarking embodied AI tasks. This makes EVA-Bench distinct from Pandora’s evaluation setup.
> ## 3. Model Difference (EVA Model vs. Pandora’s Model):
>   EVA and Pandora share the general idea of leveraging multimodal foundational models (ChatUniVi+Dynamicrafter), our approach emphasizes reasoning-driven generation, rather than direct multimodal-to-video synthesis. Therefore, our training stages and alignment targets, including loss design are different from Pandora.
>   1. **Input-Output Difference (QA-Text&Video vs. Text-Video):**
> A key distinction between EVA and Pandora lies in input and output. EVA allows Question-Video input and Answer-Video as the response. While Pandora only accepts text instructions as input, and its text output also simply copies the input instruction. This dual-modality output in EVA is crucial for complex reasoning tasks, as it allows the model to explicitly describe, explain, and refine its thought process before generating video sequences. By incorporating textual reasoning alongside video synthesis, EVA significantly enhances its ability to handle complex multi-step tasks, making it more suitable for embodied AI and decision-making applications.
>   2. **Differences in Training Data and Strategies:**
> As detailed in the Appendix, EVA employs a unique instruction-tuning approach for text-video interleaved generation. We carefully design training data that spans four meta-tasks under the RoG framework, ensuring comprehensive coverage of embodied task reasoning. During training, we first used mixed data to align the model, similar to Pandora. Then we did one more step as QA instruction tuning. The second step differs from Pandora’s data construction, which does not explicitly incorporate interleaved text-video reasoning in the same structured way.
>   3. **Video-to-Action Module**
> Pandora is a general video generation world model. While EVA also includes a base video-to-action head that could further transfer video generation to robot action (Table 5), connecting the generation model with the real world. In EVA, we can achieve end-to-end QA-to-Robot manipulation tasks, which are also far different from Pandora. （L. 450)
>
> In summary, while EVA and Pandora share a broad vision, our work introduces a novel reasoning strategy (RoG), a dedicated embodied AI benchmark (EVA-Bench), and a structured training approach tailored for reasoning-enhanced generation. These distinctions ensure that EVA provides unique contributions beyond Pandora. We include the data and benchmark format in our anonymous pages, for a more detailed and straightforward comparison.
>
> We appreciate the reviewer’s feedback and will further clarify these differences in the revised manuscript.

---

> > ### Comment · Reviewer_aEuy · 2025-04-03
> >
> > Thank you for the detailed feedback. It has addressed the main concerns. I keep my original rating of Weak Accept.

---

> > > ### Author Response · Authors · 2025-04-04
> > >
> > > Dear Reviewer aEuy,
> > >
> > > Thank you for acknowledging our rebuttal and for your thoughtful comments. Your suggestions were very helpful in clarifying our contributions, especially the comparison between EVA and Pandora.
> > >
> > > We appreciate your continued support and the maintained "Weak Accept" rating. We will reflect your feedback in the final version.
> > >
> > > Best regards,
> > > The Authors

---

### Official Review · Reviewer_wGih · 2025-03-13

**Overall Recommendation:** 3

**Summary:**

This paper introduces Reflection of Generation, a set of reasoning strategies aimed at improving video generation models for multi-step predictions and Out-of-Distribution scenarios. To support RoG, the authors propose the Embodied Video Anticipation Benchmark  to evaluate world models across diverse tasks, and they develop Embodied Video Anticipator , a model that leverages multistage training and autoregressive strategies for high-fidelity video generation. Experimental results demonstrate EVA's effectiveness in video prediction and robotics.

Update after rebuttal: Thanks for the detailed rebuttal. I appreciate the authors' efforts in addressing my concerns and helping me better understand the work. Therefore, I will maintain my original rating Weak Accept.

**Claims And Evidence:**

Yes.

**Essential References Not Discussed:**

No.

**Experimental Designs Or Analyses:**

Yes. See "Weaknesses" for issues.

**Methods And Evaluation Criteria:**

Yes.

**Other Comments Or Suggestions:**

See Weaknesses.

**Other Strengths And Weaknesses:**

Strength:
1. The proposed RoG strategy enhances video prediction by integrating intermediate reasoning steps, enabling self-correction and improved generalization in embodied scenarios.
2. The introduction of EVA-Bench provides a comprehensive and standardized evaluation framework for world models, assessing both understanding and generation capabilities across diverse in-domain and OOD embodied tasks.
3. Extensive experiments demonstrate the effectiveness of EVA in various downstream tasks.

Weakness:
1. The paper does not provide a quantitative comparison of long video generation. How does EVA mitigate the potential issue of accumulated errors in long video generation using Autoregressive Frame Extension?
2. In Table 1, why does the In-Domain experiment only fine-tune ChatUniVi, while the stronger Qwen2 model is not included in the experiment?
3. In Table 3, have the baseline methods, LLAVA-O+EVA-Gen and Qwen2+EVA-Gen, been trained? If not, the comparison may not be fair.

Minor typos:
1. Line 340: "ChatUniVi-loRA" should be corrected to "ChatUniVi-LoRA."
2. Table 3: Incorrectly bolded values.

**Questions For Authors:**

Why is the task of generating future frames considered significant? Research in simulation environments or even on real-world machines may hold greater practical relevance.

**Relation To Broader Scientific Literature:**

No.

**Theoretical Claims:**

N/A.

---

> ### Author Rebuttal · Authors · 2025-04-01
>
> # Rebuttal to Reviewer wGih
> We sincerely thank the reviewer for the thoughtful feedback and recognition of our contributions. Below, we address the key concerns raised:
>
> ## 1: Accumulated Errors in Long Video Generation
> We appreciate the concern regarding error accumulation in long-horizon video generation with autoregressive frame extension. To mitigate this, we adopt the following strategies:
> 1. Randomized Start Frame Conditioning: During training, we randomly sample starting frames to improve generalization across temporal contexts.
> 2. Contextual Inference: During inference, while we typically extend from the last frame, we also support using the last 4–8 frames as context, which enhances temporal consistency.
> 3. Hyperparameter setting: Frame Stride (FS): We empirically adjust the FS to limit the final length, reducing the risk of error propagation on very long videos>15s.
>
> Robotic video generation poses greater challenges than generic video synthesis, due to the need for object consistency, goal completion, and temporal reasoning. Our proposed RoG mechanism, by incorporating intermediate reasoning from a VLM, adaptively selects high-quality generations and provides a self-corrective feedback loop. This significantly enhances long-term coherence without requiring frame-level supervision.
>
> ## 2: Why Only Fine-Tune ChatUniVi in Table 1 Instead of Qwen2
> Thank you for this valuable observation. As shown in Table 3, ChatUniVi+EVA-Gen slightly outperforms Qwen2+EVA-Gen in terms of GCE, particularly in zero-shot settings. This motivated our decision to select ChatUniVi as the default model for fine-tuning in Table 1. Moreover, as discussed in Line 710, we aim to explore the benefits of fully pretraining a VLM before integration.In practice, ChatUniVi offers a more accessible codebase and better support for multi-stage tuning, thus making it more suitable for our current RoG-focused evaluation. This makes it more suitable for our experiments focused on architecture-level validation of RoG.
>
> We fine-tune Qwen2 directly on the EVA dataset. Without multi-stage alignment and mixture data pertaining, Qwen2-Full Parameter fine-tuning on 300k data(Qwen2-FP) or 50k data (Qwen2-FP-50k) overfits and suboptimal generalization (weak in CLIP and GPT-4o score). Therefore, we still recommend a multi-stage training strategy for stronger VLMs to fully leverage RoG’s potential.
>
> |Model|In-Domain|BLEU-1 ↑|METEOR ↑|R-L ↑|CIDEr ↑|SPICE ↑|CLIP ↓|GPT-4o ↑|
> |----|----|----|----|----|----|----|----|----|
> |ChatUniVi| × |0.0969|0.0640|0.1497|0.0427|0.0636|27.49|9.03|
> |Qwen2| × |0.2484|0.1434|0.3255|0.8914|0.2839|28.98| 29.58|
> |Qwen2-FP-50k| ✓ |0.4189|0.2424|0.5005|2.2336|0.4404|26.73|27.19|
> |Qwen2-FP| ✓ |0.4833|0.2443|0.5129|2.7301|0.5363| 26.39|24.12|
> |ChatUniVi-LoRA| ✓ | 0.3007|0.1054|0.3268|0.8245|0.2213|24.89|31.94|
> |ChatUniVi-FP| ✓ | 0.4105|0.1809|0.4416|1.9012|0.3414|25.36|38.46|
>
> ## 3: Baseline Training Status in Table 3
> Thank you for pointing this out. We clarify that the baselines in Table 3 were trained to ensure fairness:
> - Qwen2-FP + EVA-Gen, note as EVA(Qwen) was fine-tuned on 300k same downstream data as indicated in the method section.
> - EVA(Qwen)-2stages better than Qwen2: This supports our broader claim that RoG is a model-agnostic reasoning framework: different VLMs can benefit from being embedded into the RoG pipeline for robotic video generation.
> - EVA is still SOTA: multi-stage alignments are important.
>
> |Task|Model|EVAS-L ↑|EVAS-V ↑|EVA-Score ↑|
> |-----|-----|-----|-----|------|
> |**HOW-TO**|EVA(Qwen)-2Stage|88.49|71.72|80.11|
> | |Qwen2+EVA-Gen|41.54|69.34|55.44|
> | |EVA-2Stage|85.5|73.32|79.42|
> |**Next-Step**|EVA(Qwen)-2Stage|75.54|63.83|69.69|
> | |Qwen2+EVA-Gen|42.99|60.11|51.55|
> | |EVA-2Stage|73.02|64.46|68.74|
>
> ## Question: Why video generation
> It enables us to take hypothetical actions without affecting the real environment, facilitating a low trial-and-error cost.[1]
> Compared to simulators, video generation doesn’t need detailed modeling. Unlike real-world testing, it’s more scalable and cost-efficient. Therefore, it can offer a promising foundation for building scalable and generalizable robotic systems in the real world.
>
> [1] Ding J, Zhang Y, Shang Y, et al. Understanding World or Predicting Future? A Comprehensive Survey of World Models[J]. arXiv preprint arXiv:2411.14499, 2024.
>
> ## Summary
> Qwen2 is a stronger foundation model, and our supplementary results show that EVA based on Qwen2 also performs well. However, directly fine-tuning such large models can lead to catastrophic forgetting without multi-stage alignment.
> This does not affect our main claim: RoG is a model-agnostic framework that consistently improves temporal coherence and goal completion in robotic video generation. We also highlight the importance of hybrid data and staged training to fully unlock model potential.
> As more powerful VLMs emerge, we believe EVA combined with RoG will continue to advance the field of video prediction.

---

### Official Review · Reviewer_3Yfm · 2025-03-13

**Overall Recommendation:** 3

**Summary:**

This paper introduces Reflection of Generation (RoG), an intermediate reasoning strategy that enhances video prediction by combining pre-trained vision-language and video generation models as a world model. We introduce Embodied Video Anticipation Benchmark (EVA-Bench) to evaluate these models across diverse tasks and OOD settings. Based on RoG, we develop Embodied Video Anticipator (EVA), a multistage-trained model that generates high-fidelity frames and adapts to long video sequences. Experiments show EVA’s effectiveness in video generation and robotics, advancing large-scale pre-trained models for real-world applications.

**Claims And Evidence:**

The paper provides extensive experimental validation for its proposed methods, including comparisons across multiple baselines on diverse benchmarks. The introduction of EVA-Bench strengthens the evaluation by systematically assessing both in-domain and out-of-distribution (OOD) performance. However, while the empirical results demonstrate improvements, some claims about the necessity of Reflection-of-Generation (RoG) for long-horizon video prediction could be better substantiated with ablation studies isolating its impact.

**Essential References Not Discussed:**

Not found

**Experimental Designs Or Analyses:**

The experimental design is generally sound, with clear task decomposition and evaluation metrics, particularly in EVA-Bench, which assesses both understanding and generation. The comparisons against baselines are well-structured, demonstrating EVA’s advantages in long-horizon video prediction, but some evaluations, such as the role of RoG in performance gains, could benefit from stronger ablations.

**Methods And Evaluation Criteria:**

The proposed Reflection-of-Generation (RoG) approach aligns well with the challenge of making video prediction models more robust and adaptive in embodied scenarios. EVA-Bench provides a well-structured evaluation framework, covering multiple levels of task complexity, including goal completion and adaptive reasoning. While the benchmarks are relevant and diverse, additional real-world deployment results could further validate the applicability of the method beyond simulated environments.

**Other Comments Or Suggestions:**

NA

**Other Strengths And Weaknesses:**

Strengths

1. The Reflection-of-Generation mechanism is an innovative approach that introduces intermediate reasoning steps into video generation, allowing for self-correction and adaptive long-horizon forecasting. This is a meaningful departure from traditional video diffusion models, which often struggle with consistency over extended sequences.

2. The introduction of EVA-Bench provides a well-structured evaluation framework that assesses both in-domain and out-of-distribution generalization, addressing a critical gap in embodied AI evaluation. The decomposition of tasks into action description, task completion verification, and next-step prediction enhances the clarity of model assessment.

3. The EVA model demonstrates state-of-the-art performance across multiple video generation and robotics applications, showing robust transferability to real-world scenarios. By evaluating EVA on robot simulation and embodied interaction datasets, the paper strengthens its claims of applicability beyond purely synthetic settings.

Weakness

1. While the paper presents quantitative improvements, it lacks fine-grained ablation studies isolating the contribution of RoG beyond basic comparisons. A more detailed breakdown of RoG’s impact (e.g., performance with vs. without intermediate reasoning steps) would better justify its necessity.

2. Some methodological details, particularly regarding the training setup for EVA and the specific architecture choices, could be better documented for reproducibility. While EVA-Bench is a strong contribution, it would benefit from open-source code or dataset access, enabling broader validation by the research community.

**Questions For Authors:**

NA

**Relation To Broader Scientific Literature:**

The paper builds on prior work in world models and video prediction, integrating ideas from pretrained vision-language models (VLMs) and autoregressive video generation to improve long-horizon consistency.

**Theoretical Claims:**

No theoretical claims are required for review in this work.

---

> ### Author Rebuttal · Authors · 2025-04-01
>
> # Response to Reviewer 3Yfm
>
> We thank the reviewer for their thoughtful and encouraging feedback. We are glad that the reviewer appreciated the novelty of our Reflection-of-Generation (RoG) mechanism, the well-structured EVA-Bench, and the demonstrated real-world applicability of our EVA model. Below, we address the two main concerns raised point by point:
>
> ---
> ## 1. Insufficient Ablation on the Reflection-of-Generation (RoG) Mechanism
> We appreciate the reviewer’s point regarding the need for more fine-grained ablation studies to isolate the contribution of RoG. To maintain clarity in the main paper, we had streamlined ablation findings. In the revision, we will include a dedicated subsection. We will revise the experimental section to include a dedicated subsection for RoG ablations. Below, we provide a summary of the key findings:
> 1. **In the Finish-Thinking Video Generation task(Tab. 2)**, we conducted ablations by applying RoG using different vision-language models (VLMs) on the same fine-tuned generation model (EVA-Gen). The GCE scores consistently improve with RoG, even when using non-finetuned VLMs, demonstrating that RoG—by introducing intermediate reasoning—can effectively enhance task completion.
> 2. **In robotic task scenarios (Tab. 4 and 5)**, RoG-supervised generation significantly boosts task success rates:
>   - In-domain tasks: While performance of the baseline is already strong, RoG shows benefits in longer-horizon tasks such as "place into", increasing the number of successful executions (Tab. 4).
>   - Out-of-distribution (OOD) tasks: RoG leads to over 2x improvement in success cases compared to the baseline version without RoG (L.430). This supports our claim that RoG enables adaptive correction and robustness in unfamiliar scenarios.
>
> These studies support the necessity and effectiveness of the RoG mechanism. We will explicitly restructure and expand the ablation section in the revision.
>
> ---
> ## 2. Reproducibility and EVA-Bench Availability
> Thank you for pointing this out. We fully agree with the reviewer that reproducibility and accessibility are crucial for community adoption and validation. **We will open-source the code, upload a preview-version during rebuttal.**  We include part of the data and some key code in an anonymous link https://sites.google.com/view/icml-eva#h.w1phh0qv55ho We have taken concrete steps to address these concerns as follows:
> - We will organize the appendix for more clear description, and merge important details into the main paper. More  information in appendix includes: model architecture in Sec. "A. Appendix: Model Architecture and Training" L660, VGM and fine-tuning method Ensamble-LoRA in sec A.3(L.740),  training detail of VLM in Sec A.5 (L.815), and detailed model architecture and hyperparameters in L.774 and Tab.7. The dataset construction and most construction prompts are also included in Fig.11, Tab.12~18. As we claimed in L.710, we noticed that the fully fine-tunned VLM on a mixture data performs better than fine-tunned on a pretrained backbone, therefore, we choose Dynamicrafter and ChaUniVi since they have a good opensource code base with comprehensive scripts on multistage training.
>
> Regarding EVA-Bench, we commit to releasing:
> - A subset of curated data examples on our project page
> - Data processing scripts and annotations
> - Download instructions for third-party datasets used (subject to licensing constraints)
> As noted in L.817, we outlined multiple stages in the experimental pipeline, but we agree that more granularity is needed. We will expand this section to provide clearer information for the research community.
>
> ---
> We again thank the reviewer for the insightful feedback, which has helped strengthen the paper.

---

> > ### Comment · Reviewer_3Yfm · 2025-04-04
> >
> > My concerns are well addressed. Therefore I will keep my original rating and recommend this work for accepting.

---

> > > ### Author Response · Authors · 2025-04-05
> > >
> > > Dear Reviewer 3Yfm,
> > >
> > > We sincerely thank for carefully reading our response and for the constructive initial feedback. We are glad that the clarifications regarding the RoG ablation study and EVA-Bench reproducibility addressed your concerns.
> > >
> > > **We appreciate your continued recommendation for acceptance** and your recognition of the contributions of our work. Your feedback has been very helpful in improving the clarity and rigor of the paper.
> > >
> > > Thank you again for your time and support！
> > >
> > > All Authors

---

### Decision · Program_Chairs · 2025-05-01

**Decision:**

Accept (poster)

**Comment:**

This paper proposes the Reflection of Generation (RoG) method, leveraging intermediate reasoning to improve video prediction and use as a world model for multi-step prediction tasks (with a focus on robotics). The model (Embodied Video Anticipator) leverages multi-stage training and a VLM-supervised generation process that can support generation of extended video sequences. In order to test the method, EVA-Bench is proposed which includes both in and out-of-distribution videos within egocentric and robotics domains.

  Overall, reviewers appreciated the problem, method, and especially the inclusion of the EVA-Benchmark to more rigorously test this problem under both in and out-of-distribution generalization settings. Some concerns were raised, including a better demonstration/justification of the necessity of RoG for long-horizon video prediction, more detailed ablation studies, better documentation of various decisions made in the design of the architecture for reproducibility, and better situation with respect to some prior methods. The authors provided a thorough rebuttal of all of these points, and all of the reviewers stated that their major concerns were addressed.

  Considering all of the materials, I recommend acceptance of this paper. The idea is an interesting combination of VLMs and generation through reasoning mechanisms, and the benchmark has potential for increasing work in these areas in a comparable and focused manner. I strongly recommend that the authors incorporate the various discussions and paper organization feedback provided during the reviewing process to make the camera-ready version stronger.